# Orbital frontal cortex updates state-induced value change for decision-making

**Emily T Baltz[1†], Ege A Yalcinbas[1,2], Rafael Renteria[1], Christina M Gremel[1,2]\***

[1]Department of Psychology, University of California, San Diego, La Jolla, United States; [2]The Neurosciences Graduate Program, University of California, San Diego, La Jolla, United States

**Abstract** Recent hypotheses have posited that orbital frontal cortex (OFC) is important for using inferred consequences to guide behavior. Less clear is OFC's contribution to goal-directed or model-based behavior, where the decision to act is controlled by previous experience with the consequence or outcome. Investigating OFC's role in learning about changed outcomes separate from decision-making is not trivial and often the two are confounded. Here we adapted an incentive learning task to mice, where we investigated processes controlling experience-based outcome updating independent from inferred action control. We found chemogenetic OFC attenuation did not alter the ability to perceive motivational state-induced changes in outcome value but did prevent the experience-based updating of this change. Optogenetic inhibition of OFC excitatory neuron activity selectively when experiencing an outcome change disrupted the ability to update, leaving mice unable to infer the appropriate behavior. Our findings support a role for OFC in learning that controls decision-making.

DOI: https://doi.org/10.7554/eLife.35988.001

**\*For correspondence:**
cgremel@ucsd.edu

[†]These authors contributed equally to this work

**Competing interests:** The authors declare that no competing interests exist.

Decision-making depends on the ability to infer the consequences of our potential behavior. The orbital frontal cortex (OFC) has recently been hypothesized to underlie this predictive capability (*Bradfield et al., 2015*; *Schuck et al., 2016*), with OFC representing hidden state task space that functions to combine predictive information with memories of perceptually similar rewards or sensory information to control future behavior (*Wilson et al., 2014*; *Stalnaker et al., 2015*; *Schuck et al., 2016*). In support of this hypothesis, OFC appears necessary to infer outcome representations from predictive cues following a reduction in that outcome's desirability (*Gallagher et al., 1999*; *Gottfried et al., 2003*; *Izquierdo et al., 2004*; *Pickens et al., 2005*; *Takahashi et al., 2009*; *Camille et al., 2011*; *West et al., 2011*; *Jones et al., 2012*; *Rudebeck et al., 2013*). More recent work has suggested that OFC may also control model-based behavior, where goal-directed actions are controlled by the knowledge and value of the action consequence produced (*Gourley et al., 2013*; *Gremel and Costa, 2013*; *Rhodes and Murray, 2013*; *Bradfield et al., 2015*; *Gourley et al., 2016*; *Gremel et al., 2016*). Within the framework of these two behavioral controllers, one interpretation is that OFC function underlies the ability to infer that a behavior made would produce a now devalued outcome. However, what role OFC plays in perceiving the changed consequence and updating the outcome representation later used for goal-directed decision-making is not clear.

An interesting aspect to the above hypothesis is that OFC representation of task space is used to retrieve outcome representations of an expected value. In the above examples, while the current devalued outcome has to be inferred for behavioral control because it is unobservable (testing conducted in the absence of reinforcement), the subject has accrued extensive experience with the outcome's value change during prior devaluation procedures. In addition, devaluation procedures and testing are conducted in quick succession, making it difficult to separate learning processes from those controlling decision-making. Murray and colleagues (*Murray et al., 2015*) made comparisons between inhibition of OFC area 11 or 13 in non-human primate OFC prior to sensory specific

**eLife digest** As we go about our daily lives, we do not simply react to the world around us. Instead we build up mental representations of the world and use these to guide our behavior. For example, we know that if we are hungry we can go into the kitchen to get a slice of our favorite cake. But we also adapt our behavior when circumstances change. If you have just eaten an entire box of cookies you are unlikely to go looking for cake.

Which parts of the brain help us to adapt our decisions to reflect our circumstances? To find out, Baltz et al. trained mice to press levers in order to receive sugar water. In initial experiments the mice completed the training while not hungry. Afterward, some of the mice were placed on a diet that made them hungry. A re-exposure period then occurred where the mice could taste more sugar water without having to press the lever. Finally, the next day, they were given the opportunity to press the levers again.

Mice that were hungry during the re-exposure period pressed the levers more than mice that had been re-exposed while full. Further experiments showed that this was true regardless of how hungry the mice were when they first learned the task. The mice updated how much they valued the sugar water – and so changed how eagerly they tried to obtain it – based on how hungry they were during the re-exposure period.

Baltz et al. repeated the experiments, but this time blocked the activity of a brain region called the orbitofrontal cortex in the mice during the re-exposure period. This prevented the mice from updating how much they valued the sugar, and so they did not adjust their behavior accordingly. If hungry mice had performed the first training stage when they were full, they pressed the levers less often than expected after the re-exposure period. Likewise, full mice who had trained when they were hungry pressed on the lever more times, as if they were still hungry. This suggests that the orbitofrontal cortex helps to update the values that guide decision-making.

There are many disorders that can impair decision-making and prevent people from adjusting their behavior when circumstances change. These include addiction, in which affected individuals also show altered activity in their orbitofrontal cortex. This raises the possibility that in the future we may be able to treat disorders like addiction by restoring normal activity in this region of the brain.
DOI: https://doi.org/10.7554/eLife.35988.002

satiation and testing procedures versus inhibiting OFC prior to testing but after satiation. Their findings did suggest regional differences in OFC function in the devaluation of food (OFC area 13) versus the performance of a predictive stimulus discrimination task (OFC area 11), although in the former case inhibition lasted across satiation and testing procedures. However, an important feature of model-based behavior is the ability to adjust decision-making following a simple change in internal motivational state, a change that is independent from recent experiences with the outcome (i.e. increased or decreased general hunger state, not through outcome satiation). The contribution of OFC to updating internal representations controlling goal-directed actions following a state change is unknown.

One way to test the ability of OFC to infer appropriate actions following a change in motivational state is by probing incentive learning. Incentive learning tasks are useful to examine intricacies of model-based behaviors, as they separate the updating of value following a shift in motivation from inferring the proper use of the updated value for goal-directed control (*Balleine and Dickinson, 1991*; *2005*). There is evidence from neurophysiological studies to suggest that OFC may be involved in incentive learning processes. OFC BOLD activity can reflect sensory-specific state changes (*O'Doherty et al., 2000*; *Gottfried et al., 2003*), and furthermore, there is overwhelming evidence of OFC neuron encoding of economic and relative value (for review: *Stalnaker et al., 2015*). Neurophysiological studies conducted in OFC report changes in single neuron encoding of inferred action value or predictive cue value following changes in outcome value (*Critchley and Rolls, 1996*; *Padoa-Schioppa and Assad, 2008*; *Kennerley and Wallis, 2009*; *Kennerley et al., 2011*; *Gremel and Costa, 2013*; *McDannald et al., 2014*; *Stalnaker et al., 2014*; *Rich and Wallis, 2016*).

To directly examine OFC contributions to incentive learning, we adapted an instrumental task to mice, where changes in the magnitude of food restriction are used to induce motivational state changes. We show that following a state-dependent change in internal motivation (either an increase or a decrease in hunger), mice show evidence of incentive learning and subsequently use the updated values to infer appropriate decision-making. Chemogenetic attenuation and optogenetic inhibition of OFC projection neuron activity during incentive learning did not prevent increases or decreases in food palatability. However, OFC attenuation did prevent the updating of relative value changes of the food representation. Further, inhibition of OFC only during food consumption disrupted incentive learning. Our data suggests that OFC plays a critical role in updating value representations independent of valence, and these value representations are then used to foster appropriate goal-directed or model-based control over decision-making.

## Results

### State-dependent control of value updating

In order to examine state-dependent control over decision-making, we examined how changes in state alter action control independent of changes in action contingency or direct changes in outcome sensory and motivational properties. We adapted an incentive learning task previously used in rats (*Balleine and Dickinson, 1998*; *Wassum et al., 2009*) to mice. In brief, mice were trained to press a left lever under a random ratio schedule (up to RR4 or RR8) to gain access to a right lever. One subsequent right lever press (fixed ratio 1) resulted in 20% sucrose solution delivery to outcome port connected to a lickometer located in between the two levers. The left—then—right lever press chain used produces a distal seeking response relative to outcome delivery that is less sensitive to general motivational state changes than the subsequent more proximal taking response (*Balleine and Dickinson, 2005*). After acquisition, we altered the motivational state and mice were given a re-exposure period to non-contingent deliveries of the same sucrose solution. In this re-exposure phase, mice had the opportunity to experience the sucrose in a changed motivational state and undergo incentive learning. The next day, we assess whether the new motivational state induced a change in sucrose valuation by examining seeking left lever presses in a brief (5 min) non-reinforced session.

We used daily time spent under food restriction to induce different motivational states. We first trained adult (>7 weeks) male and female mice under very minimal levels of food restriction; rodent chow was removed from animals for 2 hr beginning between 1.5 and 3 hr into their light cycle (*Vollmers et al., 2009*). Immediately at the end of the 2 hr food restriction, mice were placed into the operant chamber for instrumental training. After training, mice were returned to their homecage with rodent chow readily available. Mice showed a slight increase in body weight across training (male: baseline weight = 24.55 ± 0.69 SEM, last day of training weight = 26.85 ± 0.75 SEM; paired t-test; $t_7$ = 5.33, p=0.001) (female: baseline weight = 17.79 ± 0.53, last day of training weight = 19.49 ± 0.39; paired t test, $t_7$ = 4.66, p=0.002). This suggests that the 22 hr mice had access to home-cage rodent chow was a sufficient time to maintain and increase weight. Importantly, mice had unlimited access to water except during their training sessions in the operant box.

Not surprisingly, given the minimum level of food restriction, a moderate percentage of mice failed to show evidence of lever-press acquisition for a sucrose solution. We applied a minimum response rate >0.25 left lever presses per minute (minimum 15 left lever presses in one session) on the last two days of acquisition to ensure access to right lever, and we confirmed that mice had indeed pressed the right lever for sucrose delivery. This resulted in a 31% subject loss (n = 7/22). The remaining subjects showed clear acquisition of left lever presses (last day of training average; left lever presses = 46 ± 6.5; response rate = 0.58 ± 0.08) and earned on average 4.3 ± 0.85 sucrose deliveries on the last day of training (*Figure 1—figure supplement 1A–C*).

Mice were then divided into two groups matched for response rates on the last two days of training. One group was maintained at 2 hr food restriction (Group 2–2) (n = 6), while the other group underwent a 16 hr food restriction (Group 2–16) (n = 11) (rodent chow removed 2–4 hr prior to dark cycle onset). At the end of each restriction period, mice were placed into the operant chamber where sucrose was delivered non-contingently on a random time schedule (RT120), equating to on

average one sucrose delivery every 2 min for an average of 30 sucrose deliveries across the 60 min session.

We performed lick analyses on the observed anticipatory and consummatory licking behavior (*Figure 1B–J*). Changes in licking behavior reflect palatability changes induced by the food restriction state change (*Berridge, 1991*). An increase to 16 hr food restriction produced an increase in total licks (unpaired t-test: $t_{14} = 2.39$, p=0.03) (*Figure 1B*) that was observed across the entire duration of the session (2-way repeated measures ANOVA (Group x Time block): no interaction; trend toward main effect of Group $F(1, 14)=3.28$, p=0.09; trend toward main effect of Time block $F(5, 70) =2.26$, p=0.05) (*Figure 1C*). Mice that had undergone 16 hr of food restriction trended towards being faster to initiate licking behavior ($t_{14} = 2.14$, p=0.05) (*Figure 1D*). While all mice organized their licks into bouts of licking instead of isolated licks (p>0.5) with similar inter-lick intervals within a bout (p>0.5), and similar bout durations (p>0.1), Group 2–16 contained more licks in a bout ($t_{14} = 5.05$, p=0.0002) (*Figure 1E–H*). The above analyses did not discriminate between anticipatory and consummatory licking patterns. To examine whether a motivational state change induced a different pattern of licking during sucrose consumption, we isolated the first lick burst following a sucrose delivery. In the 2–16 group, we found a trend towards an increase in burst duration following outcome delivery ($t_{14} = 1.79$, p=0.09) (*Figure 1I*), as well as a trend towards an increase in the number of licks within that first burst ($t_{14} = 1.98$, p=0.07) (*Figure 1J*). Together, these data suggest that an increase in food-restriction resulted in a state change that produced an increase in palatable licking behavior. Hence, the motivational state produced by increased food restriction increased the palatable value of sucrose.

We next examined whether the state-dependent updated incentive value was retrieved and used to control decision-making. Mice were maintained in the assigned food-restriction state and given a brief non-reinforced test session where we measured the rate of left lever pressing. State-dependent increases in sucrose value increased the outcome value as indexed by a higher response rate (*Figure 1K*). There was a trend toward Group 2–16 mice to have a higher percent of baseline response rate than Group 2–2 mice (unpaired t-test: $t_{14} = 2.10$, p=0.05). One-sample t-test against 100% to assess changes from baseline showed that mice kept at 2 hr restriction had a similar response rate to that observed on the last two days of acquisition ($t_4 = 1.19$, p>0.2). However, mice that underwent 16 hr food restriction showed an increase in response rate from baseline (one-sample t-test against 100%: $t_{10} = 3.41$, p=0.007).

To assess the contribution of context and the necessity of sucrose re-exposure to the behavioral effects observed, we performed an additional positive incentive learning experiment in naïve mice where we manipulated access to the previously trained context or sucrose during the re-exposure session (*Figure 1L*). Mice were trained on the incentive task under 2 hr food restriction. Prior to re-exposure session, all mice underwent 16 hr of food restriction, and kept at 16 hr food restriction during the test. As seen previously in rats (*Balleine et al., 1995*), the ability of the updated value to control decision-making was dependent on re-exposure to sucrose in the new motivational state, as re-exposure to the context alone (no sucrose delivered) was insufficient to change action control (*Figure 1M*). Further, the training context contributed minimally to sucrose re-exposure, as sucrose re-exposure in a novel context was sufficient to update sucrose value and increase seeking response rates (*Figure 1M*). A two-way ANOVA (Sucrose x Context) did not show a significant interaction ($F (1, 29)=0.13$, p=0.7) or a significant effect of Context ($F(1,29) = 0.8$, p=0.38). However, a main effect of Sucrose was observed ($F(1,29) = 4.96$, p=0.03). Hence, mice readily show positive incentive learning, whereby a change in action control following an increase in motivational state requires state-dependent experience to update value representations later used for goal-directed decision-making.

While we readily observed positive incentive learning in mice, it may be that an increase in value (in this case induced by an increase in hunger) exerts more control over decision-making than a decrease in value. However, outcome devaluation experiments used to probe goal-directed control suggests that mice are indeed sensitive to decreases in outcome value (e.g., *Gourley et al., 2016*; *Gremel et al., 2016*). In incentive learning, experiencing food in an altered motivational state drives the relative change in outcome value, while in outcome devaluation tests sensory-specific outcome satiation or direct aversive conditioning to the outcome representation (lithium chloride pairings with outcome) is commonly used to directly change outcome value. Effects of sensory-specific outcome devaluation are often compared to a control state where subjects are pre-fed a control

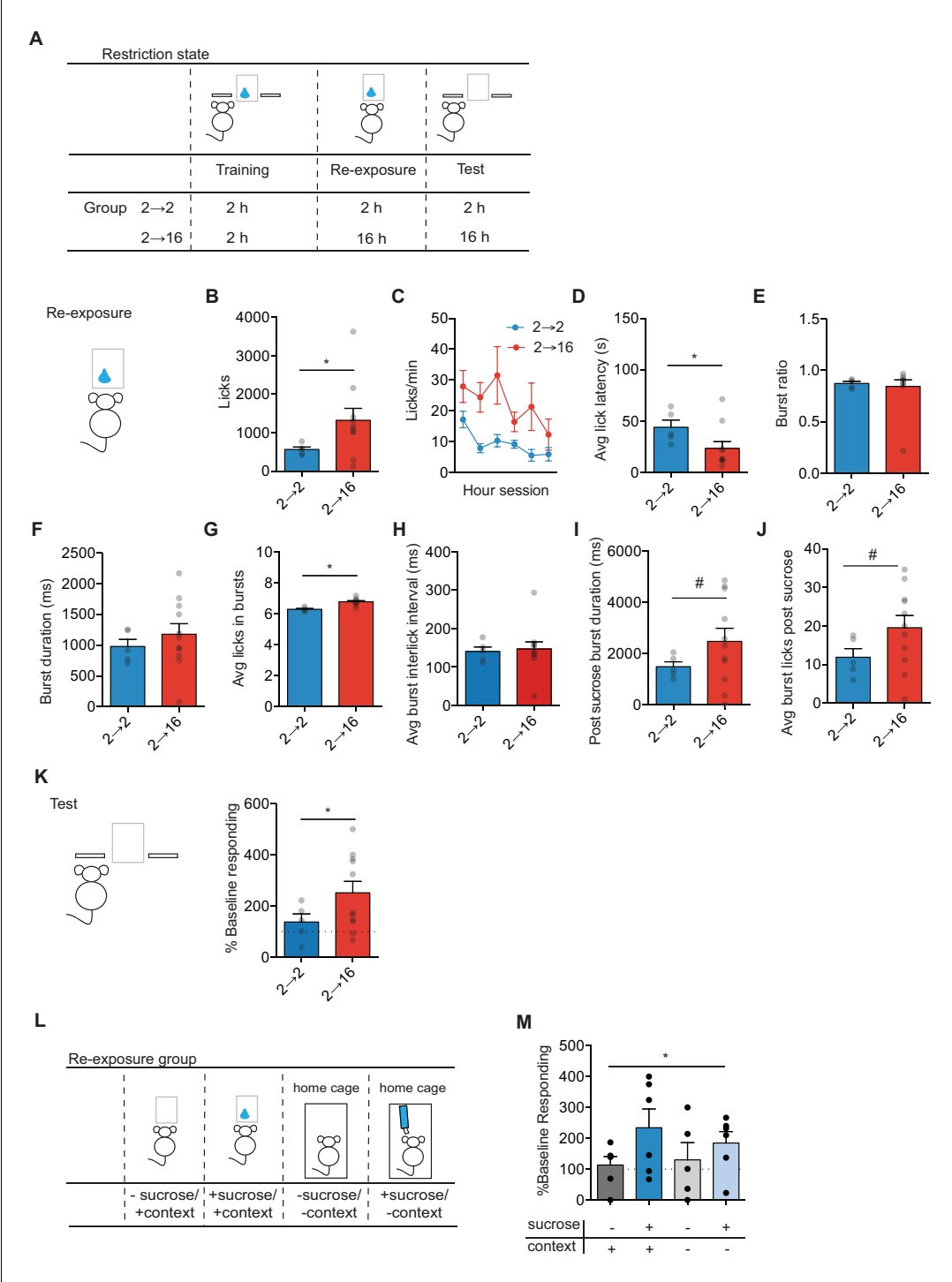

**Figure 1.** Positive incentive learning in mice. (A) Schematic showing training, re-exposure and testing schedule for positive incentive learning. Group *n*'s: 2→ 2: n = 5; 2→ 16: n = 11. Data points and bar graphs represent the mean ± SEM. (B) Number of licks and (C) licking rate (10 min bins) during the re-exposure session. (D) The average latency to begin licking after a sucrose delivery (s). (E) The ratio of licks that occur in bursts, (F) average duration of bursts (ms), (G) average number of licks within a burst, and (H) average interlick interval within bursts (ms). (I) Average burst duration after a sucrose delivery and (J) average number of licks within a burst after a sucrose delivery. (K) Response rate during the 5 min non-rewarded test as a percent of acquisition response rate (last 2 days of training). (L) Schematic of training, re-exposure, and testing schedule for context positive incentive learning. Group *n*'s: context + sucrose -: n = 5; context + sucrose + : n = 11; context - sucrose -: n = 11; context – sucrose +: n = 11 (M). (J) Percent of baseline responding (last two training days) for mice not exposed to sucrose, not exposed to sucrose nor the context, exposed to sucrose in the context, and exposed to sucrose in the home cage. * indicates p=0.05, # indicates p=0.06.

*Figure 1 continued on next page*

*Figure 1 continued*

DOI: https://doi.org/10.7554/eLife.35988.003

The following figure supplement is available for figure 1:

**Figure supplement 1.** Acquisition of lever pressing for positive and context incentive learning.

DOI: https://doi.org/10.7554/eLife.35988.004

outcome to control for general effects of satiation (e.g. *Gremel and Costa, 2013*; *Gremel et al., 2016*) and tested in a non-reinforced session without any re-exposure to the trained outcome. Incentive learning would arise if those subjects in the sated control condition (Valued day) were given a reinforced session, where subjects would lever press and experience the outcome in the sated state and undergo incentive learning (*Balleine and Dickinson, 2005*).

We next examined the capacity for negative incentive learning in mice and asked whether a state-dependent decrease in value will also alter action control. Mice underwent 16 hr daily food restriction across lever press acquisition. Food was removed from cages 3–4 hr prior to dark cycle onset, and mice were trained and tested 2–3 hr into their next light cycle. Mice were able to maintain and increase their baseline weight across acquisition (male: prior to training = 21.8 g ± 0.36; last training day = 24.5 g ± 0.51; paired t-test: $t_7$ = 14.74, p<0.0001) (female: prior to training = 15.9 ± 0.51; last training day = 18.5 g ± 0.61; paired t-test: $t_6$ = 11.02, p<0.0001). Hence, the 6–7 hr where food was present was sufficient to maintain baseline bodyweight. In contrast to positive incentive learning, mice readily acquired lever press training (only 1/17 removed for low response rate) (*Figure 2—figure supplement 1A–C*).

We then induced a change in motivational state. One group of mice was kept on 16 hr food restriction (Group 16–16) (n = 8), while for the other group food restriction was reduced to only 2 hr (Group 16–2) (n = 7) prior to the sucrose re-exposure session. The reduced duration of food restriction affected appetitive and consummatory licking behaviors measured during the re-exposure session. Reducing food restriction to 2 hr led to a decrease in the total number of licks measured (unpaired t-test: $t_{13}$ = 3.93, p=0.0017) (*Figure 2B*) that was present across the first half of the session (two-way repeated measures ANOVA (Group x Time block): interaction $F_{(5, 65)}$=6.95, p<0.001; main effect of Group $F_{(1, 13)}$=15.41, p=0.001; main effect of Time block $F_{(5, 65)}$=15.68, p<0.0001) (*Figure 2C*). Groups showed a similar latency to the first lick in a session (p>0.1) (*Figure 2D*). However, Group 16–2 mice organized fewer of their licks into bursts ($t_{13}$ = 2.82, p=0.02), and those bursts were shorter in duration ($t_{13}$ = 3.06, p=0.01), contained fewer licks ($t_{13}$ = 6.86, p=0.0002), and had longer inter-lick intervals within a burst ($t_{13}$ = 3.2, p=0.01) (*Figure 2E–H*). We next examined consummatory licking behavior tied to sucrose delivery, we found that bursts following outcome delivery were shorter in duration ($t_{13}$ = 3.29, p=0.006), and contained fewer licks ($t_{13}$ = 3.55, p=0.006) (*Figure 2I,J*). Together, this data suggests that shortening of food restriction from 16 hr to 2 hr resulted in a decreased motivational state that reduced the palatability of sucrose, with mice showing less appetitive and consummatory licking behaviors.

To examine whether the state-dependent decrease in sucrose value would be retrieved and used to control decision-making, we performed a non-reinforced test session the following day in both groups of mice. Experiencing sucrose in a decreased motivational state induced incentive learning as assessed by the decreased seeking response observed during testing (*Figure 2K*). Response rates between groups differed significantly (unpaired t-test: $t_{13}$ = 4.5, p=0.0015). Group 16–2 mice reduced their response rate from baseline (one-sample t-test against 100%: $t_6$ = 3.1, p=0.02), while Group 16–16 showed an increase from baseline (one-sample t-test against 100%: $t_7$ = 3.66, p=0.008). Hence, mice also readily show negative incentive learning where a change in action control following a decrease in motivational state requires state-dependent experience to update value representations later used for goal-directed decision-making. While positive incentive learning reflects a state-dependent increase in sucrose value, negative incentive learning reflects a relative decrease in sucrose value, with the state-dependent decrease in sucrose value supporting less seeking behavior.

## Orbitofrontal cortex activity controls state-dependent value updating

Previous work has found that OFC neurons encode action value (*Gremel and Costa, 2013*), choice value (*Rich and Wallis, 2016*), value estimates (*Padoa-Schioppa and Assad, 2006*; *Kennerley and*

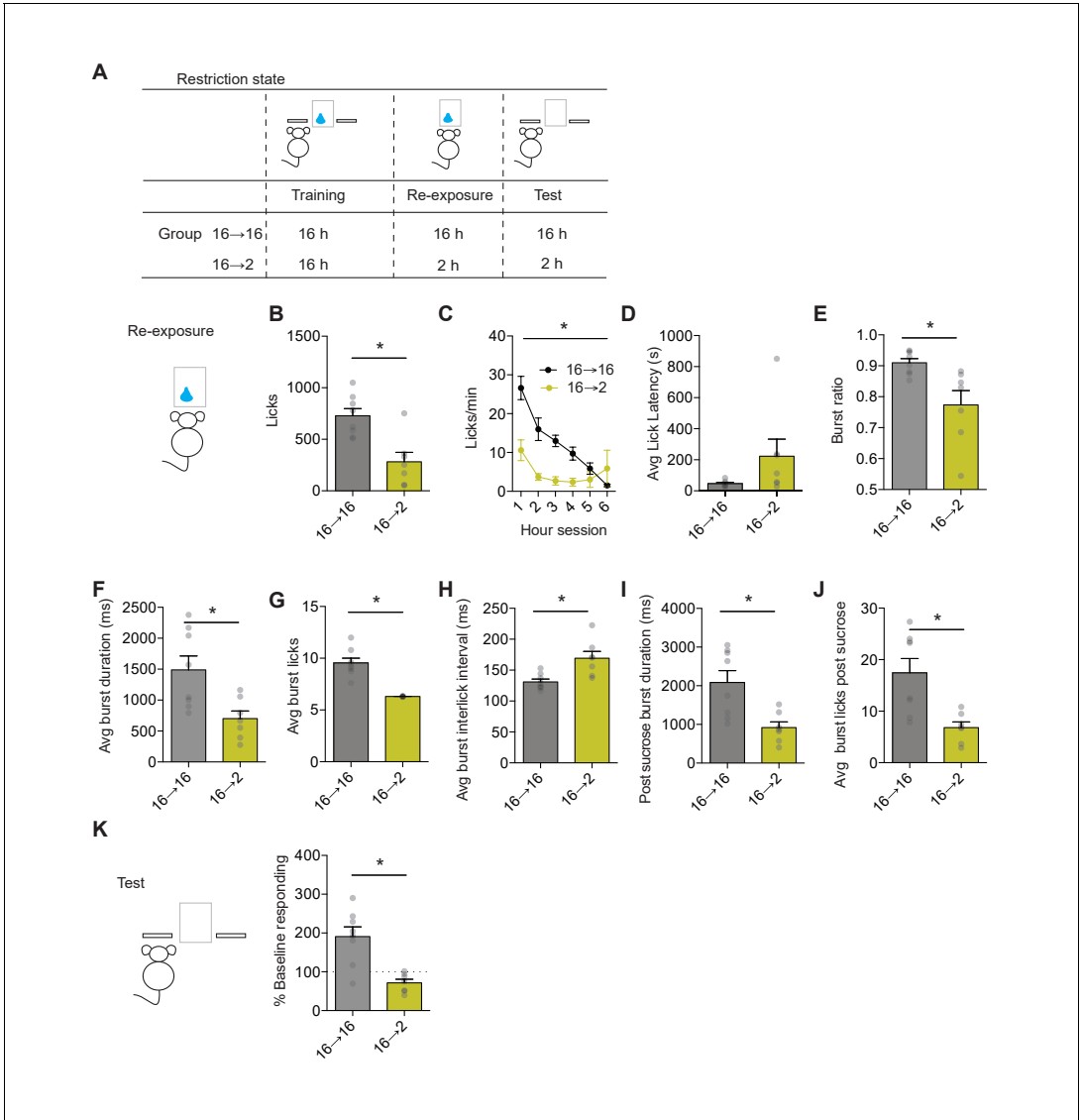

**Figure 2.** Negative incentive learning in mice. (A) Schematic showing training, re-exposure, and testing schedule for negative incentive learning. Group *n*'s: 16→ 16: n = 8; 16→ 2: n = 7. Data points and bar graphs represent the mean ± SEM. (B) Number of licks and (C) licking rate (10 min bins) during the re-exposure session. (D) The average latency to begin licking after a sucrose delivery (s). (E) The ratio of licks that occur in bursts, (F) average duration of bursts (ms), (G) average number of licks within a burst, and (H) average interlick interval within bursts (ms). (I) Average burst duration after a sucrose delivery and (J) average number of licks within a burst after a sucrose delivery. (K) Response rate during the 5 min non-rewarded test as a percent of acquisition response rate (last 2 days of training). * indicates p<0.05.

DOI: https://doi.org/10.7554/eLife.35988.005

The following figure supplement is available for figure 2:

**Figure supplement 1.** Acquisition of lever pressing for negative incentive learning.
DOI: https://doi.org/10.7554/eLife.35988.006

*Wallis, 2009*; *Padoa-Schioppa and Assad, 2008*; *Padoa-Schioppa, 2009Padoa-Schioppa, 2009*; *Kennerley et al., 2011Kennerley et al., 2011*; *McDannald et al., 2014*), and sensory attributes of value (*Rolls et al., 1989*; *Critchley and Rolls, 1996*; *Pritchard et al., 2008*; *Gremel and Costa, 2013*). These findings suggest that a state-dependent increase in palatable value of sucrose could require OFC neuron encoding. However, it could also be that OFC neuron activity is necessary for updating value representations independent of any direct representation of sucrose value or cached value representation. The sucrose re-exposure day in the incentive learning task provides a unique design with which to disambiguate these two hypotheses. The first hypothesis would predict that

OFC neuron activity is necessary to produce the increase in sucrose palatability, while the latter would predict that OFC activity during sucrose re-exposure is responsible for the updating of the increased value representations subsequently used to control decision-making during testing.

To probe whether OFC activity functionally contributes to the two above hypotheses, we took a chemogenetic approach to selectively attenuate OFC projection neuron activity and thereby disrupt OFC neuron encoding during sucrose re-exposure (*Figure 3A*). We took a rigorous approach and used two methods to restrict hM4Di receptors to OFC excitatory projection neurons. First, B6.129S2-*Emx1*^tm1(cre)Krj/J mice (Emx1-Cre) backcrossed onto C57BL/6J mice for several generations, were given bilateral lateral OFC injections of AAV5-hSyn-DIO-hM4D(Gi)-mCherry (DIO-H4) (100 nl per side; UNC Vector Core). Since the Emx1-Cre line expresses Cre-recombinase in excitatory projection neurons, this manipulation restricted DREADD expression to OFC excitatory projection neurons. Second, C57BL/6J mice were given bilateral lateral OFC injections of AAV5-CamKIIa-GFP-Cre (CamKII-Cre) (100 nl per side; UNC Vector Core) and AAV5-hSyn-DIO-hM4D(Gi)-mCherry (100 nl per side; UNC Vector Core). This also limited Cre recombinase and DREADD expression to excitatory CamKIIa-expressing projection neurons in OFC. Additional Emx1-Cre and C57BL/6J mice were given injections of AAV5-hSyn-DIO-mCherry (DIO-mCherry) or AAV5-CamKIIa-GFP-Cre (100 nl per side; UNC Vector Core) with DIO-mCherry to control for any effects of surgery, AAV infection, and CNO administration. We found no differences between strains in control measures across task acquisition (positive or negative incentive learning), re-exposure licking, or percent of baseline responding on the non-rewarded test day (see *Supplementary file 1*, Table 1) and strains were combined for the remaining analyses. To confirm function of our manipulation, we conducted whole-cell current clamp recordings in identified OFC projection neurons expressing mCherry from infusions of AAV5-hSyn-DIO-hM4Di-mCherry (*Figure 3B*). Bath application of CNO (10 μM) resulted in a significant decrease in excitability (two-way repeated measures ANOVA (Current x CNO), interaction: F (10, 70)=13.75, p<0.001; main effect of CNO and current (Fs > 11.80, ps <0.003) (*Figure 3C*, n = 8).

Following surgical procedures, all mice underwent incentive learning task training. Prior to the sucrose re-exposure session, mice were given injections of 0.9% saline (10 ml/kg) or CNO (1 mg/kg, 10 ml/kg). We used viral and drug controls in each experiment and did not see differences between controls injected with saline or CNO (*Supplementary file 1*, Table 2); therefore, we collapsed across controls for ease of presentation. Attenuating OFC activity during the sucrose re-exposure session did not alter the appetitive or consummatory licking behavior observed. Increasing food restriction led to more licks independent of OFC attenuation (2-way repeated measures ANOVA (food restriction Group x Treatment): no interaction F = 2.6, p=0.11; main effect of Group: F(1, 43)=26.21, p<0.0001; no main effect of Treatment: F = 1.13, p=0.29) (*Figure 3E*). When we examined licking rate in 10 min blocks across the 60 min session, we found that food restriction groups differed across the entire session independent of treatment (3-way repeated measures ANOVA (Session block x Treatment x Group); no 3-way interaction: F = 0.47, p>0.79; no interaction of Session block x Treatment: F = 0.72, p=0.61; interaction of Session block x Group: F (5, 225)=6.98, p<0.001; main effect of session block: F(5, 225)=21.06, p<0.001) (*Figure 3F*). In addition, the decrease in average latency to the first lick following an increase in food restriction was similar between Treatments (main effect of Group: F (1, 43)=10.88, p=0.002; no interaction or main effect of Treatment: Fs <0.5, ps >0.48) (*Figure 3G*). Food restriction groups differentially organized their licks into bursts (no interaction: F = 0.04, p=0.83; main effect of Group: F (1, 43)=5.81, p=0.02; no main effect of Treatment: F = 0.65, p=0.42) (*Figure 3H*). All mice showed similar burst durations (no interactions or main effects, Fs <4, ps >0.05) (*Figure 3I*), and the same number of licks in a burst independent of food restriction treatment (no interaction or main effects, Fs <3.3, ps >0.07) (*Figure 3J*). All groups maintained their average inter-lick intervals within a burst (no interaction or main effects, Fs <0.97, ps >0.33) (*Figure 3K*). We next examined burst licking behaviors following the delivery of sucrose to examine consumption-related licking and found no effect of OFC attenuation on consumption behaviors. OFC attenuated mice at a higher food restriction showed similar burst durations in comparison to controls groups (main effect of Group: F(1, 43)=16.46. p=0.002; no interaction or main effect of Treatment: Fs <2.8, ps >0.10), and showed a similar increase in licks within a burst following sucrose (main effect of Group: F(1, 43)=17.31, p=0.0001; no interaction or main effect of Treatment (Fs <2.92, ps >0.09) (*Figure 3L,M*). Together, our data show that OFC attenuation during re-exposure to sucrose in a changed motivational state did little to alter the increased palatability of sucrose.

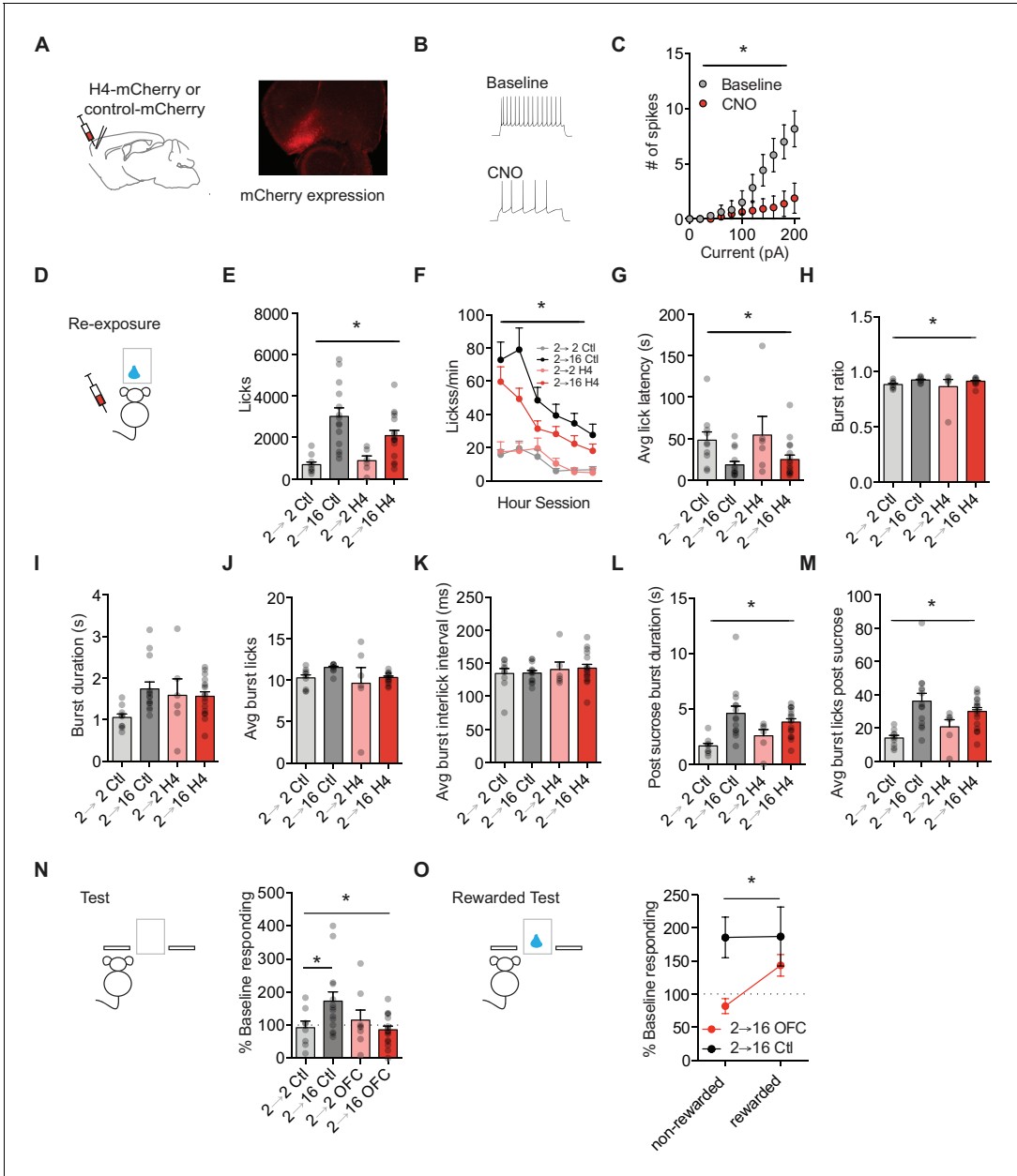

**Figure 3.** Orbitofrontal cortex attenuation prevents positive incentive learning. (A) Schematic of injection site (left) and representative mCherry expression in OFC (right). (B) Representative traces and (C) summary data from ex vivo physiological whole cell recordings in HMD4$_i$ expressing OFC projection neurons during baseline and following CNO bath application to H4 slice. (cells $n$ = 8). (D) Training and testing schematic showing when OFC manipulations were given, with CNO given only during the re-exposure session. Group $n$'s: 2→ 2 Ctl: n = 9; 2→ 16 Ctl: n = 14; 2→ 2 H4: n = 7; 2 → 16 H4: n = 17. (E) Number of licks and (F) licking rate (10 min bins) during the re-exposure session. (G) The average latency to begin licking after a sucrose delivery (s). (H) The ratio of licks that occur in bursts, (I) average duration of bursts (s), (J) average number of licks within a burst, and (K) average interlick interval within bursts (ms). (L) Average burst duration after a sucrose delivery (s) and (M) average number of licks within a burst after a sucrose delivery. (N) Response rate during the 5 min non-rewarded test as a percent of acquisition response rate (last 2 days of training). (O) Percent of baseline responding from non-rewarded to the rewarded test Data points represent individual subjects and bar graphs and error bars represent the mean ± SEM. * indicates p<0.05.

DOI: https://doi.org/10.7554/eLife.35988.007

The following figure supplements are available for figure 3:

**Figure supplement 1.** Orbitofrontal cortex excitation does not generate an increased motivational state.
DOI: https://doi.org/10.7554/eLife.35988.008

**Figure supplement 2.** Left lever presses in OFC positive incentive learning during test day.

*Figure 3 continued on next page*

Figure 3 continued

DOI: https://doi.org/10.7554/eLife.35988.009

To examine whether increased OFC projection neuron activity was necessary to update state-dependent value representations during the sucrose re-exposure, we subsequently tested all mice the next day when OFC activity was intact. Attenuating OFC projection neuron activity during sucrose re-exposure disrupted incentive learning subsequently used to infer what actions to take. A two-way ANOVA revealed a significant interaction between Food Restriction and Treatment Group (F(1, 43)=5.54, p=0.02) (no main effects, ps > 0.1) (*Figure 3N*). Post hoc Bonferroni-corrected comparisons within each Treatment found that Control 2–16 mice that underwent a state-dependent increase in motivation had significantly higher response rates than Control mice kept at 2 hr food restriction (2-2) (p<0.05). In contrast, OFC 2–16 mice that were shifted from 2 hr to 16 hr food restriction and had OFC attenuated during sucrose re-exposure showed similar response rates to mice with OFC attenuated but kept in the training motivational state (OFC 2–2) (p<0.05). Indeed, only Control 2–16 mice showed a significant increase from baseline response rates (one-sample t-test against 100%: $t_{13}$ = 2.65, p=0.02), while Control 2–2, OFC 2–2, and OFC 2–16 mice did not (ps > 0.2). This suggests that OFC projection neuron attenuation prevented the experience-based updating of state-dependent increases in sucrose value.

The lack of increased response rate during subsequent testing suggests that OFC 2–16 mice did not have an updated value representation to retrieve, and instead relied on the representation learned during acquisition to control decision-making. We then gave a subset of mice the opportunity to update sucrose value representations in a test session the next day where lever presses were rewarded (i.e. they earned sucrose deliveries). With OFC intact, OFC 2–16 mice showed increased response rate during the rewarded test compared to the non-rewarded test session ($t_{28}$ = 3.10, p=0.004) (*Figure 3O*). Control 2–16 mice had already updated the state-dependent increase in sucrose value during the re-exposure session, with their response rate consistent between non-rewarded and rewarded test sessions ($t_{22}$ = 0.02, p=0.97). Together, these data suggest that OFC projection neuron activity was necessary for mice to learn about relative increases in value which they subsequently used to infer how valuable their actions would be, but does not contribute to state-dependent changes in value perception itself.

Our above results implicate a necessity for OFC projection neuron activity for positive incentive learning. Given prior findings suggesting OFC activity positively correlates with multiple aspects of value (*Padoa-Schioppa, 2009*), one could make the hypothesis that some pattern of OFC activity is necessary for relative decreases in value. However, previous findings examining cue-outcome associations failed to find evidence of OFC neurons decreasing firing rate when delivered outcomes were less than expected (*Takahashi et al., 2009*; *2013*). It may be that OFC activity is necessary for incentive learning processes in general following a state change, be that relative increases or decreases in value.

To examine whether increases in OFC projection neuron activity are also necessary to update state-dependent decreases in sucrose value, we injected Emx1-Cre and C57BL/6J mice with the necessary combinations of DIO-H4 or CamKII-Cre and DIO-H4, or Control mice with mCherry into lateral OFC. Mice were then trained on the incentive learning task to lever press for sucrose (*Figure 4A*). Prior to the re-exposure session, mice were given an injection of CNO (1 mg/kg, 10 ml/kg) or 0.9% saline (10 ml/kg). Similar to what we observed with negative incentive learning (*Figure 2*), we found that a decrease in motivational state produced by a reduction from 16 hr to 2 hr food restriction resulted in reduced appetitive and consummatory licking behaviors. A two-way ANOVA of food restriction Group x Treatment show a similar reduction in total number of licks between Treatment groups following a decrease in food restriction (main effect of food restriction Group, F (1,63) =28.21, p<0.0001) (*Figure 4B*), that was similar across the session duration (no interaction of Session block x Treatment x food restriction Group, F = 1.76, p=0.12; no effect of Session block x Treatment, F = 1.20, p=0.31; interaction of Session block x Group, F(5,315) = 3.79, p=0.002; Main effect of Session duration, F(5,315) = 9.534, p<0.001.) (*Figure 4C*). OFC attenuated mice showed a similar latency to start licking with the re-exposure session (main effect of Group: F(1, 63)=21.79, p<0.0001; no interaction or main effect of Treatment: Fs <3.12, ps >0.08) (*Figure 4D*). OFC attenuation also

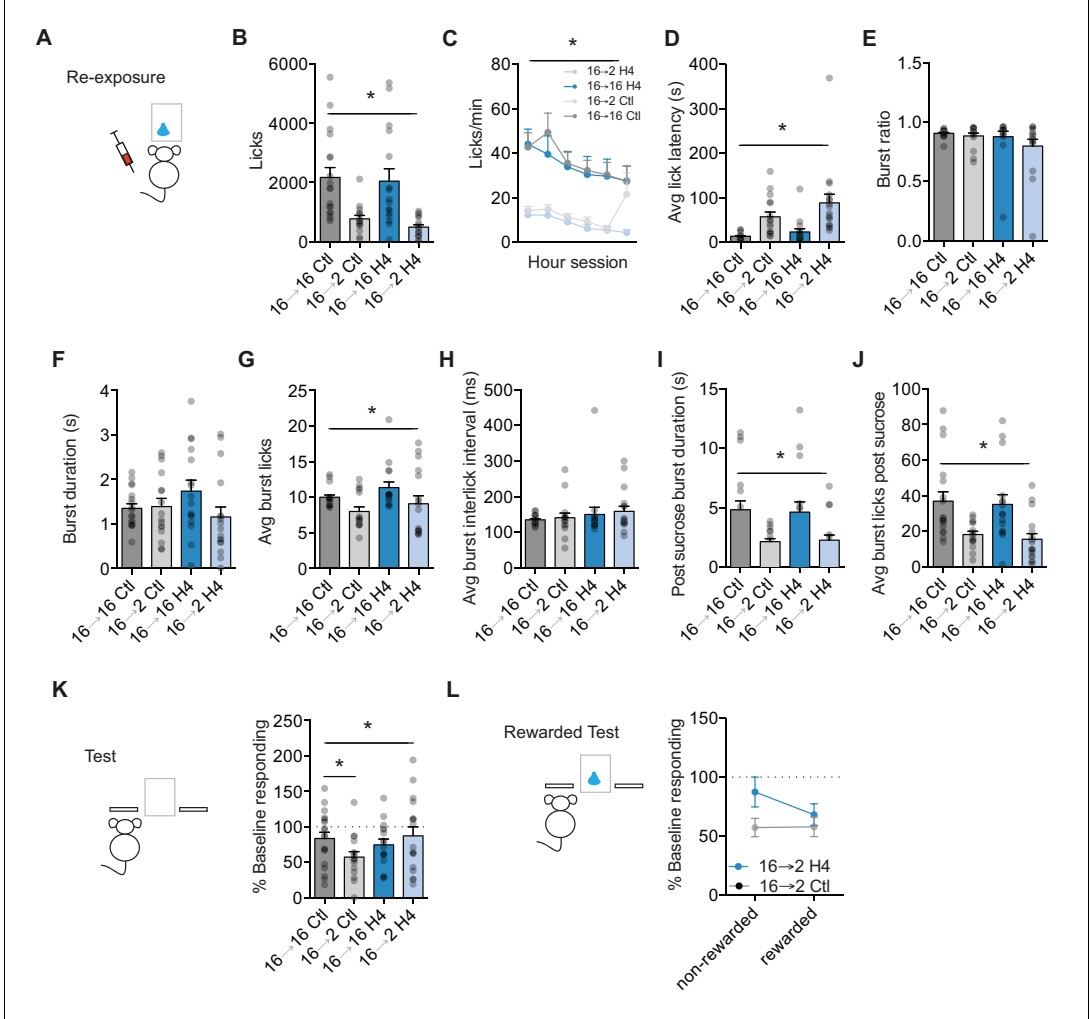

**Figure 4.** Orbital frontal cortex attenuation prevents negative incentive learning. (A) Training, re-exposure, and testing schematic showing that OFC attenuation occurred only during the re-exposure session. Group *n*'s: 16→ 16 Ctl: n = 18; 16→ 2 Ctl: n = 16; 16→ 16 H4: n = 16; 16→ 2 H4: n = 17.(B) Number of licks and (C) licking rate (10 min bins) during the re-exposure session. (D) The average latency to begin licking after a sucrose delivery (s). (E) The ratio of licks that occur in bursts, (F) average duration of bursts (ms), (G) average number of licks within a burst, and (H) average interlick interval within bursts (ms). (I) Average burst duration after a sucrose delivery (s) and (J) average number of licks within a burst after a sucrose delivery. (K) Response rate during the 5 min non-rewarded test as a percent of acquisition average response rate (last 2 days of training). (L) Percent of baseline responding from non-rewarded to the rewarded test. * indicates p<0.05.

DOI: https://doi.org/10.7554/eLife.35988.011

The following figure supplements are available for figure 4:

**Figure supplement 1.** Orbitofrontal cortex inhibition does not change sucrose preference.

DOI: https://doi.org/10.7554/eLife.35988.012

**Figure supplement 2.** Left lever presses during OFC negative incentive learning test day.

DOI: https://doi.org/10.7554/eLife.35988.013

did not affect the organization of licks into bursts (no interaction or main effects: Fs <2.2, ps >0.14), or the average burst duration (no interaction or main effects: Fs <2.57, ps >0.11) (*Figure 4E–F*). While decreases in the duration of food restriction did reduce the number of average number of licks within a burst (main effect: F(1, 63)=7.87, p=0.006), there was no effect of OFC attenuation (no interaction or main effect of Treatment: Fs <2.60, ps >0.11), nor was there any effect on the inter-lick interval within a burst (no interaction or main effects, Fs <1.48, ps >0.22) (*Figure 4G–H*). Examining potential effects of OFC attenuation on consummatory licking patterns, we examined licks within the first burst following sucrose delivery. Once again, a decrease in food restriction reduced burst

duration (main effect of Group: $F_{(1, 63)}=16.16$, p=0.0002) and the number of licks within a burst (main effect of Group: $F_{(1, 63)}=20.08$, p<0.001), but neither were altered by OFC attenuation (no interactions or main effects of Treatment: Fs <0.28, ps >0.59) (**Figure 4I–J**). Together, these data suggest that OFC attenuation during anticipatory and consummatory licking did not prevent a downshift in sucrose palatability following a decrease in motivational state.

We next examined whether OFC attenuation during re-exposure would disrupt the ability to subsequently infer the decrease in value to control decision-making. Attenuating OFC activity during sucrose re-exposure prevented the updating of a state-dependent decrease in sucrose value. A two-way ANOVA (food restriction Group x Treatment) revealed a significant interaction ($F_{(1, 63)}=4.16$, p=0.04) (**Figure 4K**). Planned comparisons within each Treatment found that Control 16–2 mice that underwent a state-dependent decrease in motivation had significantly lower response rates than Control mice kept at 16 hr food restriction (16-16) (p=0.01). In contrast, OFC 16–2 mice that were shifted from 16 hr to 2 hr food restriction and had OFC attenuated during sucrose re-exposure, showed similar response rates to mice with OFC attenuated but kept in the training motivational state (OFC 16–16) (p=0.41). Control 16–2 mice showed a significant reduction from baseline (one-sample t-test against 100%: $t_{15} = 5.53$, p<0.001) and control 16–16 mice did not (p=0.15). While OFC 16–16 and OFC 16–2 groups of mice had similar response rates during testing, OFC 16–16 mice showed a slight but significant reduction from baseline (one-sample t test against 100%: $t_{15} = 3.17$, p=0.006). Importantly, OFC attenuation in OFC 16–2 mice prevented any shift from baseline (p=0.33). Hence, attenuating OFC activity during sucrose re-exposure in a decreased motivational state prevented the updating of a decreased sucrose value. Indeed, during a subsequent reward test session with OFC intact, OFC 16–2 mice appeared to decrease responding from non-rewarded to rewarded tests and Control 16–2 mice stayed the same, although this difference was not significant (ps >0.2) (**Figure 4L**).

In addition, OFC attenuation did not alter sucrose perception. Prior to a two-bottle choice test, mice free fed for at least 4 days, were pretreated with CNO or saline for 20–30 min, and then had access to bottles of 20% sucrose and water for one hour. Both OFC inhibited and control mice were able to discriminate between 20% sucrose and water, showing a similar preference for sucrose (2-way-ANOVA, no effect of Treatment, no interaction, main effect of Sucrose concentration: $F_{(1, 36)}=24.47$, p<0.0001) (**Figure 4—figure supplement 1A**). The same cohort also underwent a 2-bottle choice test in which mice similarly discriminated 4% from 20% sucrose (2-way-ANOVA: no interaction; main effect of Sucrose concentration: $F_{(1, 36)}=11.11$, p=0.002; main effect of Treatment: $F_{(1, 36)}=8.421$, p=0.006) (**Figure 4—figure supplement 1B**). While OFC inhibited mice consumed slightly less overall during a 1 hr period (main effect of treatment), this effect was driven by one replication. Overall, these data suggest that altered sucrose perception does not contribute to the pattern of current findings following OFC inhibition.

## OFC inhibition time-locked to sucrose consumption disrupts value updating

So far, our data suggests that OFC activity is generally necessary to update value representations. However, general OFC attenuation across the entire state-task space in which mice are showing anticipatory and consummatory licking behavior in a trained context following a state change does not provide information about the temporal specificity of when OFC activity is necessary to update value changes. It could be that updated value representations accrue across time in a diffuse manner, derived from changes in anticipatory as well as consummatory behavior. Thus, OFC inhibition at any point within re-exposure session would be sufficient to disrupt value updating. However, given recent findings showing OFC encoding of sensory information (**Wikenheiser et al., 2017**) and our observation that chemogenetic activation of OFC did not increase value (**Figure 3—figure supplement 1**), OFC may use information gained during the sucrose consumption period to update relative changes in value representations. Thus, we hypothesized that OFC inhibition while consuming sucrose would disrupt value updating.

To directly examine when OFC activity is necessary for value updating, we took an optogenetic approach with the goal of inhibiting OFC projection neuron activity selectively during sucrose consumption or randomly throughout the re-exposure session. To inhibit OFC projection neurons, we relied upon a commonly used approach of expressing channelrhodopsin in parvalbumin (PV) interneurons such that light activation would induce a PV inhibitory clamp of projection neuron activity

(e.g. *Li et al., 2016*) (*Figure 5B*). B6;129P2-*Pvalb*$^{tm1(cre)Arbr}$/J (PV-Cre) mice were given an injection of AAV Ef1a-DIO-hChR2(H134R)-eYFP (ChR2) bilaterally into the OFC, and optic fiber ferrules were targeted to lateral OFC (*Figure 5A*). To inhibit OFC projection neurons selectively during sucrose consumption, we employed closed-loop feedback during the re-exposure session such that the first lick after a sucrose delivery would result in light activation of PV interneurons (delay ~10–20 ms) (*Figure 5D*). To target our light delivery to encompass the duration of sucrose consumption, we use a delivery of light at 20 Hz, 5 ms pulse width for 5 s (longer than our average burst durations following outcome delivery). We verified our ability to induce an inhibitory clamp on OFC projections ex vivo, where in whole-cell recordings we saw optogenetic activation of PV interneurons inhibited projection neuron spiking for 5 s, with no evidence of an increase in rebound firing post light activation (*Figure 5C*). Thus, we were able to selectively inhibit OFC projection neuron activity during sucrose consumption.

Mice with channelrhodopsin expressed and ferrules implanted were trained on our negative incentive-learning task. We used a modified negative incentive learning task to reduce subject loss and increase response rates. Mice were chronically food restricted to 90% of their baseline free-feeding weights throughout lever-press training. Following acquisition, mice were assigned to one of two groups that were matched for acquisition response rates. The evening before the re-exposure session, mice were given access to ad libitum food in their home-cage for their dark cycle. During the subsequent re-exposure session, our experimental group (ChR2) had light delivery paired with the first lick following sucrose delivery in the re-exposure session (*Figure 5D*). Each ChR2 mouse had a Yoked control mouse (matched for response rates during acquisition), whose light delivery was yoked to the ChR2 mice and independent of any behavior exhibited by the Yoked mouse. Thus, we had two groups with different conditions of light delivery; for one group light delivery was tied to the direct sucrose consumption experience and the other light was delivered randomly throughout the session.

Light delivery and the subsequent inhibition of OFC projection neurons had little effect on anticipatory and consummatory licking behaviors exhibited across the re-exposure session. While it appeared ChR2 mice made fewer licks overall than Yoked mice, it was not significant (unpaired t-test: $t_{16} = 1.54$, p=0.14) (*Figure 5E*). When we looked at the licking rate across the session, we did see an interaction (2-way repeated measures ANOVA (Treatment x Time): $F_{(5, 80)}=2.54$, p=0.035) and a main effect of Time ($F_{(5, 80)}=3.52$, p=0.006) but not of Treatment (p=0.14) (*Figure 5F*), suggesting that ChR2 mice may have a different pattern of licking across the session. However, there were no group differences in other lick measures as assessed through paired comparisons. There were no differences seen in the latency to make the first lick within the session (p=0.89), the organization of licks into bursts (p=0.36), the duration of bursts (p=0.18), the average licks within a burst (p=0.09), or the inter-lick interval within a burst (p=0.75) (*Figure 5G–K*). When we evaluated consumption licking behavior (i.e. after an outcome delivery), we did see an effect of light delivery on the duration of the licking burst (unpaired t-test: $t_{16} = 2.211$, p=0.041) (*Figure 5L*). However, we saw no difference in the number of licks in a burst (p=0.07) following outcome delivery (*Figure 5M*). Our data suggests that light delivery during sucrose consumption did not grossly alter licking behavior in comparison to our yoked controls. It appears OFC inhibition during consumption leaves sucrose palatability largely intact.

However, when we assessed whether mice had updated the decrease in outcome value through a non-rewarded test session the next day, we found that OFC projection neuron inhibition while mice were directly consuming sucrose appeared to disrupt value updating (*Figure 5N*). Yoked mice significantly reduced test response rates from acquisition baseline (one-sample t-test against 100% baseline: $t_7 = 3.07$, p=0.018), while ChR2 mice did less so ($t_7 = 2.20$, p=0.06). Six out of eight pairs of ChR2 mice and their yoked control showed a decreased response rate in yoked versus ChR2 mice, although the group difference was not significant (paired t-test: $t_5 = 1.67$, p=0.14). Our data suggest that OFC inhibition during sucrose experience decreases value updating. Importantly, Yoked mice showed evidence that they had updated a decrease in outcome value corresponding to experiencing sucrose in a decreased motivational state, and ChR2 mice did not. While we attempted to deliver light throughout the normal duration of sucrose consumption by targeting the first licking bout after an outcome delivery, we cannot rule out that some sucrose consumption happened outside this time frame in our ChR2 group. Further, it could be that the inhibition overlaid sucrose consumption in our Yoked group. While our data does not speak to how much sucrose experience is

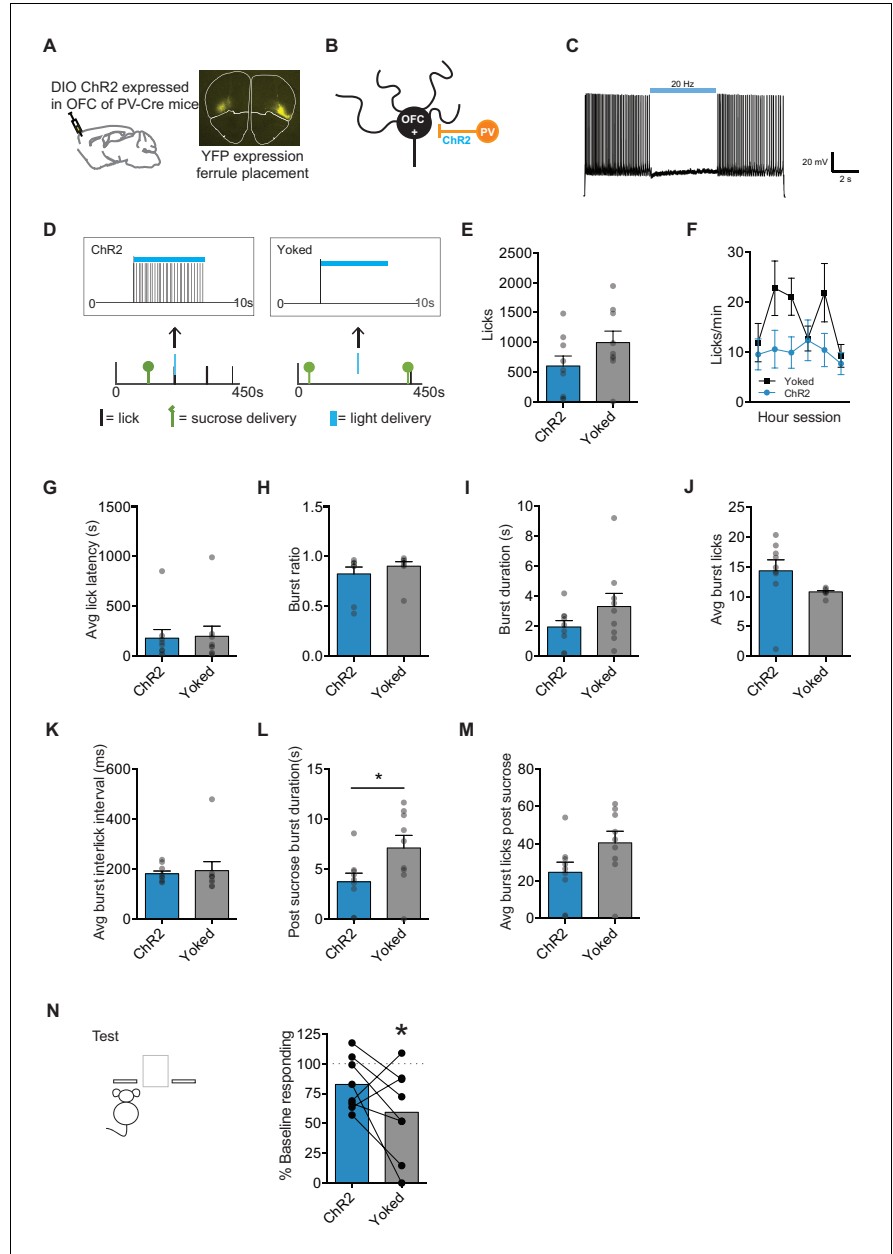

**Figure 5.** Optogenetic inhibition of OFC projection neurons during sucrose consumption prevents value updating. (A) Schematic of injection site and ferrule implant (top), with DIO ChR2-eYFP detected at OFC injection site (bottom). (B) Schematic of setup of OFC excitatory neuron (OFC +) inhibition by activation of Parvalbumin (PV) interneurons (C) Confirmation of ChR2 function using ex vivo whole-cell recording. (D) Closed-loop behavioral control over light delivery. Example where in the ChR2 group the first lick after a sucrose delivery resulted in a 5 s light delivery (5 ms pulse, 20 Hz) (left), while Yoked group received light delivery at the same time independent of licking behavior (right). Group n's: ChR2: n = 8, Yoked: n = 8. (E) Number of licks and (F) licking rate (10 min bins) during the re-exposure session. (G) The average latency to begin licking after a sucrose delivery (s). (H) The ratio of licks that occur in bursts, (I) average duration of bursts (ms), (J) average number of licks within a burst, and (K) average interlick interval within bursts (ms). (L) Average burst duration after a sucrose delivery (s) and (M) average number of licks within a burst after a sucrose delivery. (N) Response rate during the 5 min non-rewarded test as a percent of acquisition response rate (last 2 days of training). Data points and bar graphs represent the mean ± SEM. * indicates p<0.05.

DOI: https://doi.org/10.7554/eLife.35988.014

necessary to be able to update value changes, our data does suggest that OFC inhibition overlapping with sucrose experience interferes with the ability to update, while OFC inhibition not explicitly paired with sucrose consumption does not.

## Discussion

Here we find that the OFC is involved in updating value changes subsequently used to control decision-making, highlighting the contribution of OFC to model-based action control. We used an incentive learning task as a tool to differentiate the ability to learn or update value representations from deploying use of those representations. We found OFC projection neurons are necessary to learn relative value changes that are subsequently used to inform goal-directed behavior. Importantly, this is independent of the valence of the motivational state change, as OFC activity was necessary to encode both decreases and increases in relative value. The latter finding contradicts previous hypotheses suggesting a role for OFC in behavioral disinhibition. Further, our finding for the necessity of OFC activity when directly experiencing value changes suggests that OFC performs computations based on sensory information directly gained from the consumption experience. In light of OFC's prior implications in direct sensory and gustatory processing (*Grabenhorst et al., 2008*; *Lara et al., 2009*; *Ohla et al., 2012*), we hypothesize OFC updates value changes from an experienced aspect derived from sensory information.

Model-based learning requires the ability to readily update associative information then used to infer the most appropriate action. Most often, devaluation tasks have been used to examine model-based or goal-directed control, where devaluation procedures are immediately followed by tests assessing to what degree the expected outcome value controls responding. Manipulations have largely affected both devaluation and testing periods, therefore making it difficult to separate contributions of updating value from inference control during performance, or changes to action outcome contingency and motivation. In addition, compensatory brain mechanisms that arise across long durations of OFC attenuation may obscure contributing roles (*Murray et al., 2015*). Previous work has shown that optogenetic activation of lOFC only during testing selectively increases goal-directed actions (*Gremel and Costa, 2013*), while precise optogenetic OFC inhibition prevented value updating in a Pavlovian task (*Takahashi et al., 2013*). Our present data adds to this literature by showing the lOFC activity is necessary for value updating in a decision-making task characterized as model-based control. These data suggest that mechanisms within lOFC support both model-based learning as well as the ability to infer such changes to support decision-making.

We used a task where the sensory properties of the outcome, in our case a sucrose solution, stayed the same across the entire experiment. Hence, any change in value came not from changes in outcome size, identity, or flavor, but from changes to the hunger state of the mouse. In addition, the choice of whether or not to press the lever stayed the same, and there was no manipulation to the action-outcome contingency (the same lever press-outcome relationships were in place). Mice needed to infer the consequence of their potential action and adjust their action frequency appropriately. Experiencing sucrose in an increased hunger state led to an increase in the palatability of sucrose and an increase in action frequency (*Figure 1*). In contrast, experiencing sucrose in a decreased hunger state led to a decrease in sucrose palatability and decreased response rates (*Figure 2*). Thus, in both instances of incentive learning, the representation of sucrose value was updated and subsequently used to decide whether to press the lever or not. The absence of such change in control mice was apparent when a change in motivational state was not accompanied by experience (*Figure 1*). This separation of palatability from incentive learning processes provided a unique task with which to investigate OFC's hypothesized role in inferring value (*Stalnaker et al., 2015*; *Schuck et al., 2016*).

While our data supports a crucial role for OFC in updating value representations supporting model-based behavior, our data also adds to recent findings that show OFC is not necessary for perceiving changes in palatability (*Gardner et al., 2017*). Broadly attenuating OFC projection neuron activity or temporally inhibiting OFC projection neuron activity did little to change appetitive and consummatory licking behaviors (*Figures 3–5*). Mice that had not undergone a motivational state change, but had OFC attenuated during the re-exposure session, showed licking behaviors similar to mice that had OFC function intact. This suggests that mice with OFC attenuated were still able to retrieve the sucrose representation. However, it is also possible that OFC attenuation would block

retrieval of any sucrose representation independent of a motivational state change. This contrasting hypothesis suggests sucrose delivery on its own, independent of the animal's learned behavior, would elicit innate frequencies of anticipatory and consummatory behaviors subject to modulation by motivational state. While this could conceivably be another explanation for the observed palatability changes, there were no differences in the initial latency to lick between groups. The lack of a difference suggests that OFC attenuation did not affect retrieval of the sucrose representation used to initiate the first appetitive lick (*Figures 3G* and *4D*).

The present incentive learning data suggests that updating value representations under novel circumstances can be used to infer consequences for appropriate decision-making, as re-exposure happening outside of the training context was sufficient to update the representation (*Figure 1M*). This raises the question of what information gained during re-exposure is used to update value representations. We examined this question via optogenetically inhibiting OFC (*Figure 5*), where we time-locked OFC inhibition to the initial lick response following sucrose delivery in our experimental mice and compared responding to that of yoked mice given the exact same stimulation independent of their behavior. Optogenetic inhibition reduced the ability to update value in our experimental mice and not in the yoked controls. However, given the lack of a significant group difference, some caution should be taken in the interpretation. We do know that given the precise timing of the inhibition manipulation, OFC activity was intact the majority of the session. OFC inhibition did not prevent any additional motivational processes that might occur after the direct consummatory experience (e.g., lasting pleasantness) from contributing to the sucrose value representation. Further, the yoked group had the same amount of inhibition, but not tied to any behavior. Hence, it is possible that OFC inhibition in the yoked group overlapped with sucrose consumption. In addition, we cannot be 100% confident that all sucrose consumption in our experimental mice happened within the duration of OFC inhibition. There was a small but significant difference in the duration of a licking bout following sucrose delivery, which could indicate a less consumption during light delivery. Thus, our data does not speak to how much experience with the outcome is necessary to update value representations following a motivational state change. Our data does suggest that accrued aspects of the sucrose consumption experience may be used to guide value updating processes.

Here we have shown that OFC is necessary for the updating of value under both positive and negative motivational states. To our knowledge, this is the first experiment demonstrating OFC's role in retrieving and updating the value of an outcome and storing it for retrieval under a previously learned action contingency. OFC has been hypothesized to function as an unobservable task state space (*Wilson et al., 2014*; *Schuck et al., 2016*; *Bradfield et al., 2015*). Our data offer support for this hypothesis in model-based behavior. We show that in addition to the well-documented role OFC has in inferring value to control decision-making, mice also need OFC to update outcome representations. During the re-exposure session, the unobservable component is the action portion of the action-outcome association. Our data show that OFC activity is necessary to update the outcome value used for inferring the value of the action. In conclusion, these studies provide insight into the role of orbitofrontal cortex in decision-making by showing a role for OFC in updating experienced-based changes in value.

## Materials and methods

**Key resources table**

| Reagent type (species) or resource | Designation | Source or reference | Identifiers | Additional information |
|---|---|---|---|---|
| Strain, strain background (*Mus musculus*) | Emx1-Cre | The Jackson Laboratory | RRID:IMSR_JAX:005628 | maintained on a C57BL6/J background |
| Strain, strain background (*Mus musculus*) | C57Bl/6J | The Jackson Laboratory | RRID:IMSR_JAX:000664 | |
| Strain, strain background (*Mus musculus*) | PV-Cre | The Jackson Laboratory | RRID:IMSR_JAX:017320 | maintained on a C57BL6/J background |
| Strain, strain background (*Adeno-associated virus*) | AAV5-hSyn-DIO-hM4D(Gi)-mCherry | UNC Viral Vector Core; Addgene | | |

*Continued on next page*

*Continued*

| Reagent type (species) or resource | Designation | Source or reference | Identifiers | Additional information |
|---|---|---|---|---|
| Strain, strain background (*Adeno-associated virus*) | AAV5-CamKIIa-GFP-Cre | UNC Viral Vector Core | | |
| Strain, strain background (*Adeno-associated virus*) | AAV5-hSyn-DIO-hM3D(Gq)-mCherry | UNC Viral Vector Core | | |
| Strain, strain background (*Adeno-associated virus*) | AAV5-hSyn-DIO-mCherry | UNC Viral Vector Core | | |
| Strain, strain background (*Adeno-associated virus*) | AAV Ef1a-DIO-hChR2(H134R)-eYFP | UNC Viral Vector Core | | |
| Software, algorithm | MATLAB | this paper | RRID:SCR_001622 | https://github.com/gremellab/lickingstructure; copy archived at https://github.com/elifesciences-publications/lickingstructure |
| Software, algorithm | Arduino | this paper | | https://github.com/gremellab/arduinoLEDcontrol; copy archived at https://github.com/elifesciences-publications/arduinoLEDcontrol |
| Software, algorithm | GraphPad Prism 6 | GraphPad | RRID:SCR_002798 | |
| Software, algorithm | Adobe Illustrator CS6 | Adobe | RRID:SCR_010279 | |
| Software, algorithm | AxographX | Axograph, Sydney, Australia | RRID:SCR_014284 | |
| Software, algorithm | JASP | JASP Team (2018). JASP (Version 0.8.6) | RRID:SCR_015823 | |
| Chemical compound, drug | clozapine-n-oxide | NIMH Chemical Synthesis and Drug Supply Program | C-929 | |
| Chemical compound, drug | NaCl | Thermo Fisher Scientific | S271 | |
| Chemical compound, drug | NaHCO3 | Thermo Fisher Scientific | S233 | |
| Chemical compound, drug | Dextrose | Thermo Fisher Scientific | D16 | |
| Chemical compound, drug | KCl | MilliporeSigma | P9541 | |
| Chemical compound, drug | NaH2PO4 | MilliporeSigma | S3139 | |
| Chemical compound, drug | Sucrose | MilliporeSigma | S8501 | |
| Chemical compound, drug | KMeSO4 | MilliporeSigma | 83000 | |
| Chemical compound, drug | HEPES | MilliporeSigma | H4034 | |
| Chemical compound, drug | EGTA | MilliporeSigma | 3777 | |
| Chemical compound, drug | MG-ATP | MilliporeSigma | A9187 | |
| Chemical compound, drug | Tris-GTP | MilliporeSigma | G9002 | |
| Chemical compound, drug | MgSO4 | MilliporeSigma | M2643 | |
| Chemical compound, drug | CaCl2 | MilliporeSigma | 223506 | |
| Chemical compound, drug | Picrotoxin | MilliporeSigma | P1675 | |

## Animals

Male and female C57BL/6J, B6.129S2-*Emx1*[tm1(cre)Krj]/J (Emx1-Cre), and B6;129P2-*Pvalb*[tm1(cre)Arbr]/J (PV-Cre) mice were housed 2–5 per cage under a 14/10 hr light/dark cycle with access to food (Lab-diet 5015) and water ad libitum unless stated otherwise. Emx1-Cre (obtained from breeding homozygous B6.129S2-*Emx1*[tm1(cre)Krj]/J (Jackson) x C57BL/6J in house), C57BL/6J (The Jackson Laboratory, Bar Harbor, ME, USA), and PV-Cre (obtained from breeding homozygous B6;129P2-*Pvalb*[tm1(cre)Arbr]/J x C57BL/6J mice in house) mice were at least 6 weeks of age prior to intracranial injections and at least 7 weeks of age prior to behavioral training. All surgical and behavioral experiments were performed during the light portion of the cycle. The Animal Care and Use Committee of the University of California San Diego approved all experiments and experiments were conducted according to the NIH guidelines.

## Chemogenetic manipulation of OFC excitatory projection neurons

Mice received stereotaxically guided bilateral injections via Hamilton syringe into lateral OFC (coordinates from bregma: A, 2.7 mm; M/L, 1.65 mm; V, 2.65 mm). To limit our manipulations to OFC projection neurons, we used two approaches. In the first, C57BL/6J mice were given co-injections of AAV5-CamKIIa-Cre-GFP (100 nl per side; UNC Vector Core) and AAV5-hSyn-DIO-hM4D(Gi)-mCherry (100 nl per side; UNC Vector Core) in order to express inhibitory DREADD only in CamKIIa-expressing excitatory neurons. In the second approach Emx1-Cre mice, in which Cre expression is limited to projection neurons (*Gorski et al., 2002*) were given bilateral injections of AAV5-hSyn-DIO-hM4D (Gi)-mCherry (100 nl per side; UNC Vector Core). To excite OFC projection neurons, we injected AAV5-hSyn-DIO-hM3D(Gq)-mCherry into Emx1-Cre mice. Based on our previous work examining in vivo the time course to observe circuit suppression (*Gremel and Costa, 2013*) mice were given an intraperitoneal injection with either Clozapine-*n*-oxide (CNO) (10 ml/kg 1 mg/kg dose) or 0.9% isotonic saline 20–30 min prior to testing. After testing, mice were euthanized and brains extracted and fixed in 4% paraformaldehyde. Viral spread was qualified by examining in 100–150 µm thick brain slices under a macro fluorescence microscope (Olympus MVX10).

## Optogenetic excitation of OFC parvalbumin inhibitory neurons

Mice received stereotaxically guided bilateral injections via Hamilton syringe into lateral OFC (coordinates from bregma: A, 2.7 mm; M/L, 1.65 mm; V, 2.65 mm). To limit our manipulations to OFC Parvalbumin inhibitory neurons, we used a PV-Cre line (*Hippenmeyer et al., 2005*) and injected 200 nl Ef1a-DIO-hChR2(H134R)-eYFP (Chr2). Following injections, mice were trained on the instrumental task. The last 4 days of schedule training, mice were lightly anaesthetized and had optical fibers coupled to their ferrule implants using a ceramic sleeve to accustom the mice to moving with the fibers on their heads. Prior to re-exposure to sucrose, mice had the fibers coupled to their implants and were allowed 30 min to recover from effect of anesthesia. Mice received 5 ms pulses of 470 nm light at 20 Hz (1–3 mW) for 5 s triggered by their first lick after each sucrose delivery. We chose this duration of light based on our previously collected data on burst duration after a reinforce delivery. High-powered 470 nm LEDs (Thor Labs) were controlled with custom Arduino Script (https://github.com/gremellab/arduinoLEDcontrol [*Baltz and Gremel, 2017*]; copy archived at https://github.com/elifesciences-publications/arduinoLEDcontrol). Viral spread and ferrule placement were assessed under a macro fluorescence microscope (Olympus MVX10).

## Brain slice preparation

Coronal slices (250 µm thick) containing the OFC were prepared using a Pelco easiSlicer (Ted Pella Inc., Redding, CA). Mice were anesthetized by inhalation of isoflurane, and brains were rapidly removed and placed in 4°C oxygenated ACSF containing the following (in mM): 210 sucrose, 26.2 NaHCO$_3$, 1 NaH$_2$PO$_4$, 2.5 KCl, 11 dextrose, bubbled with 95% O$_2$/5% CO$_2$. Slices were transferred to an ACSF solution for incubation containing the following (in mM): 120 NaCl, 25 NaHCO$_3$, 1.23 NaH$_2$PO$_4$, 3.3 KCl, 2.4 MgCl$_2$, 1.8 CaCl$_2$, 10 dextrose. Slices were continuously bubbled with 95% O$_2$/5% CO$_2$ at pH 7.4, 32°C, and were maintained in this solution for at least 60 min prior to recording.

## Patch clamp electrophysiology

Whole-cell current clamp recordings were made in pyramidal cells of the OFC. Pyramidal cells that expressed hM4Di were identified by the fluorescent mCherry label using an Olympus BX51WI microscope mounted on a vibration isolation table and a high-power LED (LED4D067, Thor Labs). Recordings were made in ACSF containing (in mM): 120 NaCl, 25 NaHCO$_3$, 1.23 NaH$_2$PO$_4$, 3.3 KCl, 0.9 MgCl$_2$, 2.0 CaCl$_2$, and 10 dextrose, bubbled with 95% O$_2$/5% CO$_2$. ACSF was continuously perfused at a rate of 2.0 mL/min and maintained at a temperature of 32°C. Picrotoxin (50 µM) was included in the recording ACSF to block GABA$_A$ receptor-mediated synaptic currents. Recording electrodes (thin-wall glass, WPI Instruments) were made using a PC-10 puller (Narishige International, Amityville, NY) to yield resistances between 3–6 MΩ. Electrodes were filled with (in mM): 135 KMeSO$_4$, 12 NaCl, 0.5 EGTA, 10 HEPES, 2 Mg-ATP, 0.3 Tris-GTP, 260–270 mOsm (pH 7.3). Access resistance was monitored throughout the experiments. Cells in which access resistance varied more than 20% were not included in the analysis.

## Current clamp recordings

Recordings were made using a MultiClamp 700B amplifier (Molecular Devices, Union City, CA), filtered at 2 kHz, digitized at 10 kHz with Instrutech ITC-18 (HEKA Instruments, Bellmore, NY), and displayed and saved using AxographX (Axograph, Sydney, Australia). A series of fixed current injections (20 pA increments from −80 to 300 pA) were used to elicit action potential firing and the number of spikes were counted at each current step. For verification of DREADD function, current injections were done prior to (baseline) and 15–30 min after CNO (10 µM) bath application. For PV inhibition of OFC firing, ChR2 expressing PV neurons were optically stimulated using 470 nm blue light (5 ms), delivered via field illumination using a high-power LED (LED4D067, Thor Labs). Optical stimulation was done at 20 Hz for 5 s during current injections. Data from each neuron within a treatment group was combined and presented as mean ± SEM.

## Instrumental training

We adapted an incentive learning task previously used in rats (*Balleine and Dickinson, 1998*; *Wassum et al., 2009*). Mice were trained in standard operant chambers containing two levers situated around a food magazine containing a fluid well with contact lickometers and a house light on the opposite wall within sound-attenuating boxes (Med-Associates). Regular food pellets and water were freely available prior to the start of training. In brief, mice underwent either a 2 hr (positive incentive learning) or 16 hr (negative incentive learning) food restriction each day but had unlimited access to water in their home cages. Mice were trained under a chain schedule of lever presses for a sucrose delivery (20–30 µL of 20% solution per sucrose delivery). The incentive value of sucrose was manipulated during test days by maintaining, increasing or decreasing the length of food restriction and then providing a chance for sucrose revaluation. One subject in the 16–16 Ctl group in negative incentive learning was excluded for a percent of baseline response rate more than 3 SD from the norm.

### Magazine training

On the first day, mice learned to approach the food magazine (no levers present) on a random time (RT) schedule, with a sucrose outcome delivered on average every 60 s for 30 min.

### Continuous reinforcement.

The next 3 days the mice had access to the right lever and right lever presses were rewarded on a continuous reinforcement (CRF) or FR1 schedule for up to 30 sucrose deliveries or until 60–90 min had passed. Additional CRF training days were administered as needed.

### Schedule training.

Following CRF schedule training on the right lever, training continued with the introduction of the left lever into the operant chamber. The session began with left lever out and right lever retracted. A left lever press on a random ratio one (RR1) schedule produced access to the right lever. Pressing the right lever on a FR1 schedule in turn produced a sucrose outcome and then would retract. The following day, the left lever requirement was increased to RR2. The right lever was maintained on an FR1 schedule throughout training. We subsequently increased RR requirements to RR4 or RR8 for a minimum of 4 training days. We found that mice with 2 hr restriction would not exert effort necessary to maintain an RR8 schedule requirement and were kept at RR4. Response rates from the two consecutive days of training prior to testing served as the baseline response rate. Mice with a response rate of. 25 left lever presses per minute or less and mice with a a response rate > 2 SD from the mean were excluded.

### Re-exposure and testing sessions

Mice were maintained at their training food restriction duration or were shifted to a longer or shorter food restriction duration. For positive incentive learning, separate groups of mice were either maintained at 2 hr food restriction or increased to a 16 hr food restriction. For negative incentive learning, separate groups of mice were either maintained at the 16 hr food restriction or were decreased to a 2 hr food restriction. Mice were then maintained at their assigned food restriction duration for all of testing, with each re-exposure or test session conducted at the end of their assigned food-

restriction period. For the re-exposure session, mice were given re exposure to sucrose during an RT120 session for 1 hr, with sucrose delivered on average every 2 min. The next day the mice were given a 5 min non-reinforced test session where responses on the left lever after RR schedule requirements (same RR schedule as the last two days of training) would produce the right lever; however right lever presses were unreinforced. In some experiments, mice were given a 60 min rewarded session the next day. In OFC chemogenetic manipulation experiments, CNO or saline was administered 20–30 min prior to the RT session, with no pretreatment prior to the non-rewarded test or the rewarded test.

## Role of context in incentive learning

Mice were trained on a 2 hr food restriction and then switched to a 16 hr restriction prior to re-exposure and incentive learning test sessions. For these experiments, the availability and location of the sucrose during the re-exposure session was manipulated. This produced four distinct groups: 1) mice were either placed in a home cage with no access to sucrose, 2) placed in a home cage and had ad libitum access to a bottle of sucrose for 1 hr, 3) placed in the operant context for a 1 hr session but with no sucrose deliveries, or 4) placed in the operant context for a 1 hr RT120 session with sucrose deliveries. Following these re-exposure sessions, mice were given a five-minute non-reinforced test as described above.

## Two-bottle choice test

Two cohorts of OFC-H4 mice underwent 2-bottle choice tests under ad libitum food access (at least 4 days of no food restriction). Mice were given pretreatment of CNO (1 mg/kg, 10 ml/kg) or 0.9% saline 30 min prior to testing. Mice were placed in an empty cage with grating on top to hold two small test tubes of liquid. Bottle sides were counterbalanced and sham bottles were used to estimate the amount of spillage and evaporation. Bottles were measured immediately before and immediately after an hour had elapsed. In the first test, mice had access to bottles of 20% weight/volume sucrose and water for 60 min. Final group $n$'s were OFC-H4 $n$ = 20, Control $n$ = 18. For the second test, bottles of 20% sucrose and 4% sucrose were available. Final group $n$'s were OFC-H4 CNO $n$ = 19, Saline + H4 $n$=11, mCherry + CNO $n$=8. We excluded data from mice for obvious spillage of the bottles.

## Data analyses

The alpha level was set at 0.05 for all experiments. Data were analyzed using Prism 6 (GraphPad), JASP (open-source statistical package), and custom Matlab (Mathworks) scripts. Data are presented as mean ± SEM (standard error of the mean) and averaged across counterbalanced conditions. For acquisition data, effects of Treatment (Control vs. OFC hM4D$_i$), food restriction Group (2 vs. 16 hr) and Day were analyzed with repeated-measures ANOVA on lever-presses, response rates, head entries, and licking were examined.

### Re-exposure session licking analysis

Palatability was analyzed using contact lickometers (minimum bin of 10 ms). Total quantity of licks and timing of licks were measured with MEDPC (Med Associates), and custom Matlab scripts were used to analyze licking microstructure. Bursts were defined as two or more licks with an inter-lick interval under one second. This metric was chosen from previous analyses on licking microstructure within C57BL/6J mice (*Boughter et al., 2007*). To examine licking microstructure, we examined per animal total licks, licking rate over time, average latency to start licking after a sucrose delivery, burst ratio (licks occurring within bursts over total number of licks), average duration of bursts, average number of licks per burst, inter-lick interval within a burst, number of licks in the first burst following outcome delivery, duration of burst following an outcome delivery. Two-way ANOVA (Treatment x food restriction Group) or unpaired t-tests were used as appropriate to examine statistical differences. Our custom Matlab script is available on Github (https://github.com/gremellab/lickingstructure [*Baltz and Gremel, 2018*]; copy archived at https://github.com/elifesciences-publications/lickingstructure).

## Responding analysis

The average response rate from the last 2 days of training before testing was used as the baseline response rate. Unpaired t-tests or Two-way ANOVA (Treatment x food restriction Group) were used as appropriate to compare statistical differences. For unpaired t-tests, if variances were not similar, Welch's correction was used. One-way t-tests against 100% were used to examine changes from baseline. Planned comparisons between non-rewarded and rewarded test percent baseline responding were performed to examine hypothesized presence of value updating.

## Acknowledgements

The Gene Therapy and Vector Core at the University of North Carolina provided the DREADD viruses. This research was supported by R00AA021780, R01AA026077, Whitehall and Brain and Behavior Foundations to C.MG. We thank undergraduate research assistants Ni (Jenny) Zhen, Esra Elhendy, and Roxana Demehri for the technical assistance and Roy Jungay for animal care assistance.

## Additional information

### Funding

| Funder | Grant reference number | Author |
| --- | --- | --- |
| National Institute on Alcohol Abuse and Alcoholism | R00AA021780 | Christina M Gremel |
| Whitehall Foundation | | Christina M Gremel |
| Brain and Behavior Research Foundation | | Christina M Gremel |
| National Institute on Alcohol Abuse and Alcoholism | R01AA026077 | Christina M Gremel |

The funders had no role in study design, data collection and interpretation, or the decision to submit the work for publication.

### Author contributions

Emily T Baltz, Data curation, Formal analysis, Investigation, Methodology, Writing—original draft, Writing—review and editing; Ege A Yalcinbas, Investigation, Writing—review and editing; Rafael Renteria, Investigation, Methodology, Writing—review and editing; Christina M Gremel, Conceptualization, Resources, Data curation, Formal analysis, Supervision, Funding acquisition, Investigation, Methodology, Writing—original draft, Project administration, Writing—review and editing

### Author ORCIDs

Emily T Baltz http://orcid.org/0000-0001-9770-3666
Ege A Yalcinbas http://orcid.org/0000-0002-9480-7192
Christina M Gremel http://orcid.org/0000-0002-8710-0543

### Ethics

Animal experimentation: This study was performed in strict accordance with the recommendations in the Guide for the Care and Use of Laboratory Animals of the National Institutes of Health. All of the animals were handled according to approved institutional animal care and use committee (IACUC) protocols (#S15060) of the University of California, San Diego.

### Decision letter and Author response

Decision letter https://doi.org/10.7554/eLife.35988.017
Author response https://doi.org/10.7554/eLife.35988.018

## Additional files

### Supplementary files

• Supplementary file 1. Table 1: Effects of Strain on Responses Table 2: Comparison of Saline vs. CNO-treated Controls.
DOI: https://doi.org/10.7554/eLife.35988.010

• Transparent reporting form
DOI: https://doi.org/10.7554/eLife.35988.015

### Data availability

All data generated or analysed during this study are included in the manuscript and supporting files. Code used in these studies has been deposited in Github.

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
