## [Decision Letter]

Thank you for submitting your article "Orbital frontal cortex updates motivational state-induced changes in value to control decision-making" for consideration by *eLife*. Your article has been reviewed by three peer reviewers, one of whom, Geoffrey Schoenbaum (Reviewer #1), is a member of our Board of Reviewing Editors, and the evaluation has been overseen by Michael Frank as the Senior Editor.

The reviewers have discussed the reviews with one another and the Reviewing Editor has drafted this decision to help you prepare a revised submission.

Summary:

In this study, the authors show that chemo- and opto-genetically targeted inhibition of the OFC impairs the ability of rats to learn the new value of a reward for the purposes of modifying instrumental responding. This so-called "incentive learning" is assessed by inactivating during preexposure to a food reward after an up- or down-shift in general motivational state (hunger). Inactivation of OFC during this preexposure had no effect on motivational changes in consumption but abolished later changes in a pre-trained instrumental response to obtain that food. The results illustrate a role for OFC in learning generally and in the acquisition of model-based information more specifically, including information that is necessary for normal instrumental responding.

Essential revisions:

While the reviewers generally thought the results were novel and represented an impressive set of studies, there were some concerns over the framing and interpretation and also in some of the analyses. The below represent the critical changes, though we will leave the individual reviews as well, which should be addressed as possible.

1) The framing and interpretation of the study was difficult to parse. This issue is expressed by two of the reviews, so we do not belabor it here. Basically the framing/interpretation did not seem to cleanly capture the novel aspects of the research. The reviews provide some suggestions, but of course this is up to the authors. As it stands, I think not making this clear would affect the success of the paper in a broader audience.

2) There was some agreement that moving Figure 1J from supplemental would help the uninitiated understand the significance of the incentive learning phenomenon. This seems like an important control to have represented at the outset of the paper.

3) Procedurally it was felt that collapsing the two strains of mice without providing more detail is problematic. There is a statement that there were "no differences between strains and approaches". This needs to be accompanied with some statistical evidence that there are no main effects or interactions of the factor (strain) with the critical measures. Possibly the two groups could be plotted separately in the supplemental. Depending on the power this might be sufficient to satisfy this concern.

4) There is a similar concern over the mixing of CNO and saline in the control groups. Given recent evidence that CNO is converted to clozapine and thus may have its own effects, it is important to establish clearly that CNO alone is not responsible for the effects reported, since these might be general and not OFC specific. This could be done a variety of ways. We leave it to the authors, but this needs to be addressed.

5) Generally groups were relatively small. This led to some p values being borderline or even non-significant. In one place a p = 0.05 is stated as significant. Is the threshold p<0.05 or equal? More importantly for the optogenetic experiment, the key a priori comparison is not nearly significant. We appreciate the comparisons of each group to baseline, but the comparison across groups really is the key question. In this case, we think this needs to be improved by replicating the study or perhaps removed.

For other points, please see the reviews below.

*Reviewer #1:*

In an impressive series of experiments, Gremel and colleagues demonstrate that the OFC is required for bidirectional changes in instrumental behavior after motivational changes for the predicted reward. Using chemo and optogenetic approaches they show that inactivating OFC in mice during revaluation of a reward – so-called incentive learning – disrupts later changes in lever pressing for that reward. Overall the behavior is extremely well done and rigorous in its design and analysis and the results seem very clear and important to me.

They are important for several reasons I think. First they clearly show a role for the OFC in goal- or model-based instrumental behavior. While Gremel and Costa showed this previously, it is somewhat at odds with negative data (mostly Ostlund and Balleine), so this is important by itself. Further, the results suggest the possibility that the discrepancy with prior reports could reflect a unique role for the OFC in updating the value of the sensory properties of the reward. I do not know for sure, but it seems to me that depending on procedures and the timing of your manipulations (and assuming no role for executing the instrumental action later for OFC, something not shown here), you might get variable results. Third I think this may be the first evidence that OFC is generally necessary for changes in behavior as opposed to just when things get worse; such deficits are often described as "disinhibition". I like the current result because it cannot be explained like this. Finally, this result demonstrates a clear effect of OFC on learning in the context of an inference-based behavior. This final novel bit adds to prior evidence that OFC is important for learning, but shows that the learning influenced by OFC includes (though may not be limited to) model-based or what I think of as "real" associative learning (as opposed to model-free or cached value structures). This is important to me personally since the learning effects of OFC output seem to influence dopaminergic error signals. Showing that OFC dependent learning contributes to model-based behavior later fits well with recent expansion of the types of learning dopamine errors may support. For all these reasons, I think this is an excellent study.

That is my case for the paper. On the con side, I found I had some trouble understanding fully the authors framing. I basically never felt that the question was clearly defined – indeed I am not sure that the authors question corresponds cleanly to any of the novel aspects of the results highlighted above. The authors seem to want to contrast "updating" or a role in learning versus one in just perceiving or perhaps using value. I think if the authors could sharpen their question – whether it corresponds to any of the answers I highlighted above – it would help me.

To assist, I've outlined a bit of what confused me:

For starters, I think there is good evidence already that the OFC cannot be necessary just for perceiving value. There is a long list of value based behaviors, including sucrose consumption, but also extending to things like discrimination learning, Pavlovian conditioning, and even economic choice, that are not dependent on OFC. So I was confused that a major goal of the paper seemed to be to rule this out….. I imagine the authors mean something much more specific or I misunderstood?

I was also confused by the emphasis on learning. There are a number of reports highlighting a role for OFC in learning. These range from not very well-controlled studies of reversal learning to the demonstration that inhibition of OFC during over-expectation impairs "updating" of the associative meaning of the cues. I see the unique importance of what has been done here (outlined above), but it is not clear to me whether there is something especially unique about incentive learning the authors mean to highlight or whether it can be an example consistent with these other reports.

Finally I feel like the idea of updating is not mutually exclusive with a role for the OFC in later contributing to the deployment of the information. In my opinion, OFC is clearly necessary for deploying information (distinct from any role in learning/updating) in Pavlovian behaviors. It is not clear whether it is also true for instrumental – the authors would be more expert than me in that – but the current expt does not comment on this. I was never clear if the authors meant to cite their data as evidence against any role for OFC in using information? Compounding my confusion, there are a couple studies that have tried to get at the question of OFC and updating, but they were not clearly discussed. So I am not sure how the authors think their current data add to or perhaps contradict or are different from these approaches.

Importantly these are mostly framing and interpretive issues. The experiments are beautiful, and I have no major arguments over the data really. I generally think authors have a right to discuss their data however they think best. So I guess my comments are mostly meant in that spirit. Here are a few citations that came to mind, supporting the above comments:

West et al., 2011. "Transient inactivation of orbitofrontal cortex blocks reinforcer devaluation in macaques." Journal of Neuroscience 31: 15128-15135. shows in mks that inactivation disrupts devaluation effects during selective satiation. also applies after though…. different from amygdala

Murray et al., 2015. "Specialized areas for value updating and goal selection in the primate orbitofrontal cortex." *eLife* 10.7554/*eLife*.11695.001. suggest a dissocation with area 13 mediating value updating in satiation and area 11 seemingly the use of the information later

Gardner et al., 2017. "Lateral orbitofrontal inactivation dissociates devaluation-sensitive behavior and economic choice." Neuron 96: 1192-1203. shows pretty clearly that ofc is not generally necessary for perceiving value…..

Takahashi et al., 2013. "Neural estimates of imagined outcomes in the orbitofrontal cortex drive behavior and learning." Neuron 80: 507-518. example of updating that is impaired by temporally specific optogenetic inactivation of the ofc

*Reviewer #2:*

This study adapts an incentive learning paradigm for mice in order to study the role of OFC in motivated responding. The paradigm uses different satiety states to change the value of a sucrose solution. The authors found that chemogenetic or optogenetic down-regulation of OFC projection neurons resulted in a selective impairment in incentive learning, as measured by the mouse's ability to update values that inform future behavior, when the update is based on sucrose exposure in a new motivational state.

Overall I think this study is very well conceived and executed, and the results are important. They build on earlier work showing that rodent OFC is required to update values by adding specificity in two domains: (1) they demonstrate learning-related effects of OFC manipulations with no effects on reward perception or palatability, and (2) they demonstrate a specific temporal requirement for OFC involvement during ongoing behavior. The manuscript is clearly written and overall I have no major concerns.

[Minor comments not shown.]

*Reviewer #3:*

This paper by Baltz and colleagues describes the application of an incentive learning procedure to mice, and reports the effects of DREAD and optogenetic inactivation of the OFC during the incentive learning period to test whether it is necessary for the revaluation process. The authors find positive evidence for an OFC contribution during food intake after a motivational shift, as seen one day later in weak modulation of responding for that reinforce in an extinction test. The question of the contributions of OFC to incentive learning is an important one, and relates to overall hypotheses on its role in economic choice, value encoding, and state space construction. There is much to like in this paper from the characterization of the behavioral model to using two different approaches to decrease activity in the OFC. I especially like the use of optogenetic activation of inhibitory interneurons to inhibit OFC and to circumvent issues related to inhibitory opsins. This was clearly a lot of work and there are some interesting findings here. However, I am not clear on the specific advance relative to other papers showing OFC is required during the reinforcer exposure for devaluation via sensory specific satiety (i.e., Murrey et al., *eLife* 2015). In addition, when considering experimental design, the number of mice in many groups are small, which likely contributes to some of the statistical "trends" and inconsistencies across experiments. Additional specific points below.

1) The first sentence of the Discussion states "here we uncover a cortical area that controls experience-based value updating". Can the authors please expand on how the current findings move beyond prior knowledge? For example, the Introduction does not spell out what the authors might consider are the critical distinctions between devaluation and incentive learning and the OFC's contribution to each. The discussion is equally confusing in this regard. In a related point regarding placing this work within the framework of prior studies, can the authors mention in the discussion how they see the current findings relating to the demonstrated role of the OFC in using information on changed value at test in multiple settings?

2) Many of the groups are n's of 5-7. This is quite low for mouse behavior, especially given the inter-individual variation seen in the acquisition data provided in the supplement. To firmly establish a new variant of a behavioral model, it would be ideal to run sufficient control subjects to get a firm idea of the typical behavior for both positive and negative incentive learning. What is responding during training and during test for a larger group.

3) In describing the analysis of licking behavior during incentive learning the authors analyze all licks, and then break out just those during actual consumption. Were there any effects on the "anticipatory" licks? Also in all of the different experiments, it is not clear if the amount of sucrose consumed across conditions differed. For example, did mice in the negative incentive contrast experiment drink less sucrose after a reduction in hunger than their controls?

4) There are problems with the statistics. The authors report p values of.09 and.07 as trends and a value of.05 as significant. To my mind the comparison is significant or it is not, and to be significant it should be less than 0.05. In addition, the p-vlaues just mentioned appear to be different than in the corresponding Figure 1, I and J. Later when p-values appear that are between.1 and.05 the authors do appropriately consider them non-significant. In general, if the authors suspect there may be a group difference that they are not detecting, I would suggest it may be due to their relatively low sample sizes.

5) In a related point, in some experiments, the relevant control groups fail to show the predicted effect, as seen in Control 2-16 mice wherein the increase from baseline responding was not significant for the initial DREADD experiment, and for the final optogenetics experiment in which there was not a significant group difference. Thus the authors cannot draw strong conclusions in either of these experiments.

6) There are multiple aspects of the data that should be shown that currently are not. For example, the main test data are shown as percentage of baseline responding. While this is appropriate, the actual responding should also be made available. Second, in multiple cases the authors combine mice across groups, for example, combing two kinds of transgenic mice, or combining both saline and CNO treatments in control groups. While this may also be fine, if supported statistically, a table showing the actual group data, and the number of cases in each group, for each group is required. The combination of mouse lines is potentially inappropriate given their different backgrounds (B6/129 vs B6) as these lines typically have very different instrumental behavior. In addition, given the current controversies over off-target actions of CNO, these data must be shown independently and ideally compared with saline.

7) Because the authors conducted their OFC manipulations in the same context as the later test, they cannot claim at this point that OFC inactivation is required for instances of incentive learning that occur outside that context. While this is very likely the case, the authors did not show that here.

8) Finally could the authors clarify their text in the discussion on prediction error and the paragraph on retrieval of sucrose representations. The logic is not clear/I find many of the sentences a bit inscrutable.

---

## [Author Response]

Essential revisions:While the reviewers generally thought the results were novel and represented an impressive set of studies, there were some concerns over the framing and interpretation and also in some of the analyses. The below represent the critical changes, though we will leave the individual reviews as well, which should be addressed as possible.1) The framing and interpretation of the study was difficult to parse. This issue is expressed by two of the reviews, so we do not belabor it here. Basically the framing/interpretation did not seem to cleanly capture the novel aspects of the research. The reviews provide some suggestions, but of course this is up to the authors. As it stands, I think not making this clear would affect the success of the paper in a broader audience.

We thank the reviewers for their perspectives on how this body of work fits into the field. We have changed much of the Introduction and Discussion to reflect the novelty of results found here. Specifically, we now emphasize our finding that OFC is necessary for model-based learning in that it is required for the ability to readily update associative information used for decision-making.

2) There was some agreement that moving Figure 1J from supplemental would help the uninitiated understand the significance of the incentive learning phenomenon. This seems like an important control to have represented at the outset of the paper.

We have moved Figure 1J into the main body of the paper to ensure that readers understand the importance of sucrose re-exposure in incentive learning process and the lack of contextual influence on updating. It is now seen in Figure 1M, and Figure 1L is a schematic of experimental design for that experiment.

3) Procedurally it was felt that collapsing the two strains of mice without providing more detail is problematic. There is a statement that there were "no differences between strains and approaches". This needs to be accompanied with some statistical evidence that there are no main effects or interactions of the factor (strain) with the critical measures. Possibly the two groups could be plotted separately in the supplemental. Depending on the power this might be sufficient to satisfy this concern.

We have provided statistical evidence of no strain differences Supplemental Table 1. We ran ANOVAs (Food restriction x Strain x OFC Treatment) from task acquisition (response rate), sucrose re-exposure (licking rate), and incentive learning (% of baseline response rate). We see no main effects of strain, nor any interactions of strain. Importantly, we have added text noting that the Emx1Cre line has been backcrossed to C57Bl/6 for many generations.

4) There is a similar concern over the mixing of CNO and saline in the control groups. Given recent evidence that CNO is converted to clozapine and thus may have its own effects, it is important to establish clearly that CNO alone is not responsible for the effects reported, since these might be general and not OFC specific. This could be done a variety of ways. We leave it to the authors, but this needs to be addressed.

We have provided statistical evidence of no effect of CNO vs. saline treatment in controls in Supplemental Table 2 in Supplement file 1 (licking rate and percent of baseline responding). We believe we have the right controls for CNO effects (by giving it to control animals) and do not see any effect of CNO treatment on our behaviors. Further, we would like to point to recent papers suggesting that the concern for clozapine effects is > 2h (Gomez et al., 2017; Mahler and Aston-Jones, 2018), and all of our testing is performed under 2 h.

5) Generally groups were relatively small. This led to some p values being borderline or even non-significant. In one place a p = 0.05 is stated as significant. Is the threshold p<0.05 or equal? More importantly for the optogenetic experiment, the key a priori comparison is not nearly significant. We appreciate the comparisons of each group to baseline, but the comparison across groups really is the key question. In this case, we think this needs to be improved by replicating the study or perhaps removed.

We have done additional replications and increased our n’s in our context incentive learning study, OFC positive incentive learning study, and optogenetic inhibition of OFC during sucrose consumption study. Our findings still stand. We have revised the text to reflect the new statistics. In addition, we have now clearly set criteria for statistical significance being less than 0.05 (p < 0.05).

Responses to the separate reviews follow:

Reviewer #1:

[…] For all these reasons, I think this is an excellent study.That is my case for the paper. On the con side, I found I had some trouble understanding fully the authors framing. I basically never felt that the question was clearly defined – indeed I am not sure that the authors question corresponds cleanly to any of the novel aspects of the results highlighted above. The authors seem to want to contrast "updating" or a role in learning versus one in just perceiving or perhaps using value. I think if the authors could sharpen their question – whether it corresponds to any of the answers I highlighted above – it would help me.

We thank the reviewer for their input. We have substantially changed our Introduction and Discussion, reframing our text to emphasize that OFC is necessary for model-based learning, in that it is required for the ability to readily update associative information used for decision-making. Our original emphasis had to do with OFC supporting this very goal-directed process of inferring a simple shift in motivation to learn about value change, as well as the finding of a clear separation of value perception and updating processes. We now take a step back, and we emphasize the demonstration of OFC involvement in learning used to control inference-based behavior.

To assist, I've outlined a bit of what confused me:For starters, I think there is good evidence already that the OFC cannot be necessary just for perceiving value. There is a long list of value based behaviors, including sucrose consumption, but also extending to things like discrimination learning, Pavlovian conditioning, and even economic choice, that are not dependent on OFC. So I was confused that a major goal of the paper seemed to be to rule this out….. I imagine the authors mean something much more specific or I misunderstood?

We have clarified that our finding clearly separated the separation of perceiving value change and updating value change, since our data shows perception is intact, adding to previous findings. The following text has been added to the Discussion, as well as text changes throughout to reflect this.

“While our data supports a crucial role for OFC in updating value representations supporting model-based behavior, our data also adds findings that show OFC is not necessary for perceiving changes in palatability (e.g., Gardner et al., 2017).”

I was also confused by the emphasis on learning. There are a number of reports highlighting a role for OFC in learning. These range from not very well-controlled studies of reversal learning to the demonstration that inhibition of OFC during over-expectation impairs "updating" of the associative meaning of the cues. I see the unique importance of what has been done here (outlined above), but it is not clear to me whether there is something especially unique about incentive learning the authors mean to highlight or whether it can be an example consistent with these other reports.

We have added text to the Introduction and Discussion to hopefully clarify our question. Incentive learning is a good model to examine learning in a model-based framework. For example see below:

“However, an important feature of model-based behavior is the ability to adjust decision-making following a simple change in internal motivational state, a change that is independent from recent experiences with the outcome (i.e. increased or decreased general hunger state, not through outcome satiation). The contribution of OFC to updating internal representations controlling goal-directed actions following a state change is unknown.”

Finally I feel like the idea of updating is not mutually exclusive with a role for the OFC in later contributing to the deployment of the information. In my opinion, OFC is clearly necessary for deploying information (distinct from any role in learning/updating) in Pavlovian behaviors. It is not clear whether it is also true for instrumental – the authors would be more expert than me in that – but the current expt does not comment on this. I was never clear if the authors meant to cite their data as evidence against any role for OFC in using information? Compounding my confusion, there are a couple studies that have tried to get at the question of OFC and updating, but they were not clearly discussed. So I am not sure how the authors think their current data add to or perhaps contradict or are different from these approaches.

We have clarified our manuscript to emphasize that in addition to OFC contribution to inference based behavior, our data supports an additional role for OFC in learning within a model-based framework. We agree that data supports a role for OFC in using information to control behavior. Our past work (Gremel & Costa, 2013) selectively light activated OFC excitatory neurons during testing, but following sensory-specific satiation procedure. We found that stimulation selectively increased the frequency of actions that were goal-directed (i.e. mice pressed the lever more, but only when the behavior was goal-directed).

Importantly these are mostly framing and interpretive issues. The experiments are beautiful, and I have no major arguments over the data really. I generally think authors have a right to discuss their data however they think best. So I guess my comments are mostly meant in that spirit. Here are a few citations that came to mind, supporting the above comments:West et al., 2011. "Transient inactivation of orbitofrontal cortex blocks reinforcer devaluation in macaques." Journal of Neuroscience 31: 15128-15135. shows in mks that inactivation disrupts devaluation effects during selective satiation. also applies after though…. different from amygdalaMurray et al., 2015. "Specialized areas for value updating and goal selection in the primate orbitofrontal cortex." eLife 10.7554/eLife.11695.001. suggest a dissocation with area 13 mediating value updating in satiation and area 11 seemingly the use of the information laterGardner et al., 2017. "Lateral orbitofrontal inactivation dissociates devaluation-sensitive behavior and economic choice." Neuron 96: 1192-1203. shows pretty clearly that ofc is not generally necessary for perceiving value…..Takahashi et al., 2013. "Neural estimates of imagined outcomes in the orbitofrontal cortex drive behavior and learning." Neuron 80: 507-518. example of updating that is impaired by temporally specific optogenetic inactivation of the ofc

We have added additional discussion concerning these previous findings. In particular, we add discussion about how our findings relate to those of Murray et al., 2015.

Reviewer #3:

[…] Additional specific points below.1) The first sentence of the Discussion states "here we uncover a cortical area that controls experience-based value updating". Can the authors please expand on how the current findings move beyond prior knowledge? For example, the Introduction does not spell out what the authors might consider are the critical distinctions between devaluation and incentive learning and the OFC's contribution to each. The discussion is equally confusing in this regard. In a related point regarding placing this work within the framework of prior studies, can the authors mention in the discussion how they see the current findings relating to the demonstrated role of the OFC in using information on changed value at test in multiple settings?

We have added additional text to the Introduction and Discussion to highlight how the present findings differ from previous works done, and what new information can be gained from these studies. For example, the following text has been added to the Introduction

“In addition, devaluation procedures and testing are conducted in quick succession, making it difficult to separate learning processes from those controlling decision-making. Murray and colleagues (Murray et al., 2015) made comparisons between inhibition of OFC area 11 or 13 in non-human primate OFC prior to sensory specific satiation and testing procedures versus inhibiting OFC prior to testing but after satiation. […] Incentive learning tasks are useful to examine intricacies of model-based behaviors, as they separate the updating of value change following a shift in motivation from inferring the proper use of the updated value for goal-directed control (Balleine and Dickinson, 1991; 2005).”

We have also made extensive changes to the discussion. For example, the following text has been added to the Discussion.

“Model-based learning requires the ability to readily update associative information then used to infer decision-making. […] Thus, our data suggests that mechanisms within lOFC support both model-based learning as well as the ability to infer such changes to support model-based decision-making.”

2) Many of the groups are n's of 5-7. This is quite low for mouse behavior, especially given the inter-individual variation seen in the acquisition data provided in the supplement. To firmly establish a new variant of a behavioral model, it would be ideal to run sufficient control subjects to get a firm idea of the typical behavior for both positive and negative incentive learning. What is responding during training and during test for a larger group.

We have done additional replications and increased our n’s in our context incentive learning study, OFC positive incentive learning study, and optogenetic inhibition of OFC during sucrose consumption study. Our findings still stand. We have revised the text to reflect the new statistics.

3) In describing the analysis of licking behavior during incentive learning the authors analyze all licks, and then break out just those during actual consumption. Were there any effects on the "anticipatory" licks? Also in all of the different experiments, it is not clear if the amount of sucrose consumed across conditions differed. For example, did mice in the negative incentive contrast experiment drink less sucrose after a reduction in hunger than their controls?

As anticipatory lick behavior for a given measurement is Total licking – Consummatory Licking, we did not directly examine anticipatory licking. We did a check on these analyses; there does not appear to be differences besides food restriction level in anticipatory licking. We have not included this data in the present manuscript because it seems redundant (given the anticipatory + consummatory = total). As for the second part of the question as to whether sucrose consumed differed across group, we cannot measure consumption outside of licking behavior. Further, it is hard to extrapolate that if they licked less they consumed less. This is because we don’t know how many licks are absolutely necessary to consume the outcome delivery, as this would likely differ depending on motivational state and lick efficiency. It could be the difference in licking we observe across food restriction levels is due to a motivation-induced increase in licking beyond what is needed for actual consumption. We did address the potential that OFC inhibition or CNO administration may change sucrose consumption even if the licking patterns look the same. We provided data from two bottle choice tests looking at the consumption of 20% sucrose under OFC inhibition. We saw no differences in how mice perceived and consumed sucrose as compared to water in these preference tests.

4) There are problems with the statistics. The authors report p values of.09 and.07 as trends and a value of.05 as significant. To my mind the comparison is significant or it is not, and to be significant it should be less than 0.05. In addition, the p-values just mentioned appear to be different than in the corresponding Figure 1, I and J. Later when p-values appear that are between.1 and.05 the authors do appropriately consider them non-significant. In general, if the authors suspect there may be a group difference that they are not detecting, I would suggest it may be due to their relatively low sample sizes.

We have now clearly set criteria for statistical significance being less than 0.05 (p < 0.05). We have increased our n’s in our context incentive learning study, OFC positive incentive learning study, and optogenetic inhibition of OFC during sucrose consumption study. We have revised the text to reflect the new statistics.

5) In a related point, in some experiments, the relevant control groups fail to show the predicted effect, as seen in Control 2-16 mice wherein the increase from baseline responding was not significant for the initial DREADD experiment, and for the final optogenetics experiment in which there was not a significant group difference. Thus the authors cannot draw strong conclusions in either of these experiments.

We have toned down our conclusions. We performed an additional replicate for the OFC manipulation during positive learning increasing our n, and do find a significant difference in Control 2-16 mice from baseline. We agree that in positive incentive learning experiments, often the effect does not appear as strong as those observed in negative incentive learning. However, we do see it across multiple experiments (Figure 1L and M). We attribute that to the temporary and mild increase in hunger state in the positive condition. During training, mice are not hungry, and indeed we do loose around 30% of subjects for not performed. The increase to 16 h is not a terrible restriction, as mice chronically held at 16 h food restriction maintain baseline weight. We think that mice aren’t really much more hungry during the 16 h restriction, but enough so that it often influences responding, just not to the largest degree. Indeed, when we were first piloting out the positive incentive learning task, we tried to two days of 16 h food restriction to see if we could drive responding up more during the non-rewarded test. Our responding was at similar levels, suggesting that the 16 h food restriction produces a mild hunger state.

We have added additional n to the optogenetic experiments, and we see the same pattern of findings. We have also added text to the Results and Discussion to limit our conclusions from the outcome of this study. Text from the Discussion is below:

“This raises the question of what information gained during re-exposure is used to update value representations. […] Our data does suggest that accrued aspects of the sucrose consumption experience may be used to guide value updating processes.”

6) There are multiple aspects of the data that should be shown that currently are not. For example, the main test data are shown as percentage of baseline responding. While this is appropriate, the actual responding should also be made available. Second, in multiple cases the authors combine mice across groups, for example, combing two kinds of transgenic mice, or combining both saline and CNO treatments in control groups. While this may also be fine, if supported statistically, a table showing the actual group data, and the number of cases in each group, for each group is required. The combination of mouse lines is potentially inappropriate given their different backgrounds (B6/129 vs B6) as these lines typically have very different instrumental behavior. In addition, given the current controversies over off-target actions of CNO, these data must be shown independently and ideally compared with saline.

We have included average lever presses made during testing for positive and negative incentive learning (Figure 1—figure supplement 1) and for OFC positive and negative incentive learning (Figure 3—figure supplement 2). We do agree though, that the appropriate way to show the data in the main paper is by normalizing to response rate. By doing so, it acknowledges the self-paced behavior we are examining, as mice show variation in the number of lever presses made.

7) Because the authors conducted their OFC manipulations in the same context as the later test, they cannot claim at this point that OFC inactivation is required for instances of incentive learning that occur outside that context. While this is very likely the case, the authors did not show that here.

We agree with this assessment and have tried to make clear that we are not claiming this.

8) Finally could the authors clarify their text in the discussion on prediction error and the paragraph on retrieval of sucrose representations. The logic is not clear/I find many of the sentences a bit inscrutable.

The discussion has been edited for clarity, and the paragraph about prediction error has been removed.